# Minimal Variance Model Aggregation: A principled, non-intrusive, and versatile integration of black box models

**Théo Bourdais** [†*]
California Institute of Technology
theo.bourdais@caltech.edu

**Houman Owhadi** [†]
California Institute of Technology
owhadi@caltech.edu

## Abstract

Whether deterministic or stochastic, models can be viewed as functions designed to approximate a specific quantity of interest. We introduce Minimal Empirical Variance Aggregation (MEVA), a data-driven framework that integrates predictions from various models, enhancing overall accuracy by leveraging the individual strengths of each. This non-intrusive, model-agnostic approach treats the contributing models as black boxes and accommodates outputs from diverse methodologies, including machine learning algorithms and traditional numerical solvers. We advocate for a point-wise linear aggregation process and consider two methods for optimizing this aggregate: Minimal Error Aggregation (MEA), which minimizes the prediction error, and Minimal Variance Aggregation (MVA), which focuses on reducing variance. We prove a theorem showing that MVA can be more robustly estimated from data than MEA, making MEVA superior to Minimal Empirical Error Aggregation (MEEA). Unlike MEEA, which interpolates target values directly, MEVA formulates aggregation as an error estimation problem, which can be performed using any backbone learning paradigm. We demonstrate the versatility and effectiveness of our framework across various applications, including data science and partial differential equations, illustrating its ability to significantly enhance both robustness and accuracy.

## 1 Introduction

Many challenges in scientific computing and Machine Learning (ML) involve approximating functions using models. These models can vary significantly, ranging from numerical solvers that integrate differential equations to various ML methods and other approximation techniques. Their performance can differ widely across different benchmarks, as seen in the Imagenet leaderboards (Russakovsky et al., 2015). In some cases, no single model outperforms others across all scenarios; each has its own strengths and weaknesses. For instance, the Intergovernmental Panel on Climate Change (IPCC) report on model evaluation (Flato et al., 2014) highlights numerous evaluation metrics, showing that different models excel in different areas. When multiple models are available to predict the same quantity of interest, a key challenge arises: how can their predictions be combined most effectively? Ensembling ML models is a well-established practice with many successful applications, such as bagging, boosting, and Mixture of Experts (MoE) (Zhang & Ma, 2012). Gradient boosting, in particular, is widely regarded for its effectiveness, combining simplicity and accuracy in data analysis (Bentéjac et al., 2021). Most ensembling methods focus on training strategies that build and combine multiple models (Zhang & Ma, 2012). However, methods for aggregating predictions from existing models are less developed and broadly used, and practitioners often rely on simplistic techniques like averaging (Ebert, 2001). This highlights the need for a simple and general framework to aggregate predictions and better exploit diverse models' unique strengths.

---

[*]Corresponding author.
[†]Department of Computing and Mathematical Sciences, California Institute of Technology, Pasadena, CA 91125, USA.

**Related work**  Ensemble learning methods combine multiple learnable models to improve accuracy. While these works are influential training paradigms, our approach assumes the models are not trainable, and focuses solely on their aggregation, yielding a different task. We emphasize that our work is not comparable to popular ensembles such as Gradient Boosting (Schapire, 2003) as it combines black box models.

Extensive research has been made on model averaging, especially in cases where the models are assumed to be linear (Zhang & Liu, 2023; Hansen, 2007; Liang et al., 2024). Combination strategies with constant factors include Viana et al. (2009); Pawlikowski & Chorowska (2020). Other studies have explored the pointwise linear aggregation of surrogate models, with various heuristics using weighted averages. Zerpa et al. (2005) uses the variance predicted by Gaussian processes, Lee & Choi (2014); Liang et al. (2023); Ye et al. (2020) try to predict the cross-validation error at each point and use it to combine models while Liu et al. (2016) trains an aggregation to minimize the Mean Squared Error of the aggregate. Crucially, these methods are heuristics, which do not justify why the aggregation should be a convex combination, or why it should be weighted using cross-validation. Secondly, they do not formulate the aggregation as a learning problem and instead use specific approaches, such as K-nearest neighbours or interpolation through weighted averages. This limits the scope of the aggregation, in particular, these studies only aggregate one-dimensional regression, and prevent the methods from benefiting from advances in other areas of Machine Learning.

A growing trend in scientific computing involves applying statistical inference and machine learning (ML) methods (Raissi et al., 2019; Batlle et al., 2024; Chen et al., 2021; Bourdais et al., 2024). These problems are often framed as operator learning tasks, which can be seen as infinite-dimensional regression problems. In this work, we adopt a statistical/ML framework to unify and aggregate diverse methods, even those not inherently inference or ML-based. This approach is particularly well-suited to structured problems with inherent design constraints, where heterogeneous models naturally arise. Such contexts often involve a combination of non-trainable numerical solvers, legacy simulation codes, and specialized ML models with distinct trade-offs, which are challenging to integrate or retrain using standard ensemble techniques. Our work focuses on addressing these challenges by reconciling and leveraging the strengths of pre-trained or specialized models in constraint-heavy, structured problems.

**Contributions**  We introduce Minimal Empirical Variance Aggregation (MEVA), a versatile method for aggregating predictions from diverse models, treating them as black-box input-output functions (deterministic or stochastic). By focusing on variance minimization, MEVA provides a robust, non-intrusive framework, enhancing reliability in data-scarce or noisy environments where traditional methods often fall short. While the approach is broadly applicable, its primary strength lies in addressing challenges in scientific computing and other structured domains. Key contributions include:

- **Justification and formulation**: Contrary to previous heuristics, we present a probabilistic model for aggregation, grounded in clear assumptions that justify using convex combinations and approximations of cross-validation error (Sections 2, 3). Future works may build upon this model and adapt it to constraints specific to the application at hand. We also formulate MEVA as a learning problem (sec. 3), making an extension to other settings such as graphs or computer vision simple using popular learning techniques of these fields.

- **Ruling out MEEA** We assess Minimal Empirical Error Aggregation (MEEA), which minimizes aggregate empirical error directly, and identify its limitations via a pathological example (Section 2.2). MEVA, introduced in Section 3, addresses these challenges and consistently outperforms MEEA in experiments. We rigorously prove that MEVA converges faster than MEEA, making it more robust in scarce data regime.

- **Extension to operator learning** Our approach extends model aggregation beyond machine learning to include scientific computing, enabling the integration of diverse methods for estimating shared target quantities. We demonstrate this by expanding from one-dimensional problems to operator learning using established techniques (Section 4.2). To our knowledge, this is the first work to aggregate outputs across machine learning and classical PDE-solving methods.

The remainder of the paper is structured as follows: Section 2 discusses pointwise aggregation and MEEA's limitations. Section 3 introduces the probabilistic foundation of MEVA, a theorem proving

its superiority, and its computation. Section 4 validates MEVA with two experiments: a sanity check in data science and examples from Scientific Machine Learning (SciML) involving PDE solving.

## 2 THE MINIMAL ERROR AGGREGATION

### 2.1 BEST LINEAR AGGREGATE WHEN MODEL CORRELATIONS ARE KNOWN

An effective way to derive an aggregation method is to define it as optimal with respect to a specific loss function. If the target function $Y(x)$ and the models $M_i(x)$ are viewed as random variables on the same probability space, a commonly used loss function is the Mean Square Error (MSE). In this context, an aggregated model $M_A(x)$ can be defined as any measurable function of the models that minimizes the expected squared error:

$$M_A(x) := \underset{f \text{ measurable}}{\arg\min} \mathbb{E}\left[(Y(x) - f(M_1(x), .., M_n(x)))^2\right] = \mathbb{E}\left[Y(x) | M_1(x), .., M_n(x)\right], \quad (1)$$

where the second equality follows from the $L_2$ characterization of conditional expectation. While computing the conditional expectation can be intractable (Ackerman et al., 2017), this computation reduces to solving a linear system in the Gaussian case. Specifically, if the vector $(Y(x), M_1(x), \ldots, M_n(x))$ is Gaussian with mean 0 (see remark below), then the conditional expectation in the equation above is a linear combination of the models' outputs. Rullière et al. (2017) uses this assumption to aggregate Gaussian processes. Bajgiran & Owhadi (2021) demonstrates that any consistent and rational aggregation of models must be a weighted average that is point-wise linear. Finally, many ensembling methods (random forest (Breiman, 2001), MoE (Yuksel et al., 2012)) also use a linear combination of models. Motivated by this, we restrict our aggregation approach to a pointwise linear combination of the models:

$$M_A(x) = \alpha^*(x)^T M(x) \quad (2)$$

With this simplification, aggregation can be achieved by determining the Minimal Error Aggregation (MEA) $\alpha^*(x)$ such that:

$$\alpha^*(x) = \underset{\alpha \in \mathbb{R}^n}{\arg\min} \mathbb{E}\left[(Y(x) - \alpha^T M(x))^2\right] = \mathbb{E}\left[M(x)M(x)^T\right]^{-1} \mathbb{E}\left[M(x)Y(x)\right] \quad (3)$$

By definition, the MEA is the best possible point-wise linear aggregation. This equation reveals that the MEA requires knowledge of both the correlation matrix of the models, $\mathbb{E}\left[M(x)M(x)^T\right]$, as well as the correlation between the models and the target $\mathbb{E}\left[Y(x)M(x)\right]$. We demonstrate in appendix B.2 that the aggregated model can achieve exceptional performance when these correlations are perfectly known.

**Remark** In the general case with non-zero mean, the aggregation of a Gaussian vector would be affine. Since we do not assume anything on the models, affine coefficients can be recovered by adding an extra constant model, i.e., by a linear aggregation of $\tilde{M}(x) = (M(x), 1)$. In section 2.2, we will see that this can lead to catastrophic failure for empirical error minimization. Appendix F shows that empirical variance minimization provides a more robust approach for estimating bias.

### 2.2 DATA-DRIVEN AGGREGATION

We will now consider the situation where the quantities $\mathbb{E}\left[Y(x)M(x)\right]$ and $\mathbb{E}\left[M(x)M(x)^T\right]$ required for the best linear aggregate equation 3 may be ill-defined (e.g., because the underlying models are not stochastic) or difficult to estimate. In this situation, a natural alternative approach is to estimate the aggregation coefficients $\alpha^*$ appearing in equation 2 directly from data by a minimizer $\hat{\alpha}$ of a (possibly regularized) empirical version of the loss (3), as seen in Liu et al. (2016):

$$\hat{\alpha}_E = \underset{a \in \mathcal{H}}{\arg\min} \sum_{i=1}^{N} \left|Y^i - a(X^i)^T M(X^i)\right|^2 + \lambda \|a\|_{\mathcal{H}}^2, \quad (4)$$

where the $(X^i, Y^i := Y(X^i))$ are $N$ data points; $\mathcal{H}$ is a set of functions used for the approximation; $\lambda \geq 0$ and $\|a\|_{\mathcal{H}}^2$ is a regularizing norm. We call this aggregation Minimal Empirical Error Aggregation (MEEA). While this transition from expected loss to empirical loss may seem well-founded, it can introduce a significant loss of information, which yields to the aggregation completely failing, as showcased in the following sections.

**Pathological example: A dubious trend** In this example, we introduce a basic aggregation with linear coefficients, where the MEEA fails. This, along with appendix C and G, gives intuition as to why MEEA is not the appropriate strategy. Figure 1 represents the results of this experiment. We pick $Y$, a good model $M_G$ and bad models $M_B$ s.t.:

$$\begin{cases} Y(x) & = & 2x + \cos(3\pi x) \\ M_G(x) & = & Y(x) + \epsilon(x) \\ M_B(x) & = & 1 \end{cases} \quad \text{where } \epsilon(x) \sim \mathcal{N}(0, 0.2) \tag{5}$$

We also choose a linear aggregation, which means that we will find $a_G, a_B, b_G, b_B$ s.t. $\hat{\alpha}(x) = (a_G x + b_G, a_B x + b_B) \in \mathbb{R}^2$. Finally, we pick some trick data $X = (0.8 - 2/3 * 4, 0.8 - 2/3 * 3, .., 0.8, -0.8, -0.8 + 2/3 * 1, .., -0.8 + 2/3 * 4)$ so that the $Y(X^i) = Y^i$ form a line. We may see that $M_G$ has a lower error than $M_B$ on each $X^i$ by evaluating the models at these data points. Thus, we expect an aggregation method to use the good model primarily in the aggregation. However, the opposite behaviour is observed. Minimizing the loss (4) with $\lambda = 0$, we obtain that the coefficient on the good model is 0, so $M_G$ is completely ignored, which is counterintuitive. The aggregate uses the bad model, $M_B$, to directly estimate $Y$ by linear regression. This shows that the MEEA simply employs models as features to interpolate the data points instead of weighting each model according to its estimated accuracy.

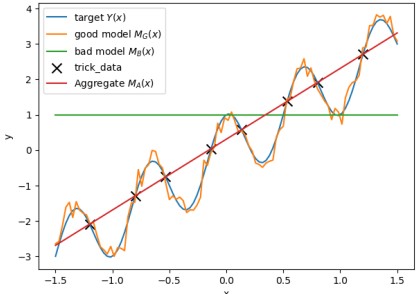

Figure 1: Pathological example (sec. 2.2)

# 3 AGGREGATION THROUGH ERROR ESTIMATION: THE MINIMAL VARIANCE AGGREGATE (MVA)

## 3.1 MODELLING THE ERROR

By placing no assumption on our model, MEEA is vulnerable to pathological examples, and in our experiments does not generalize well. In particular, there is nothing representing the fact that models are approximations of the target. To encode this, we add the assumption that the models, represented in the vector $M$, are unbiased estimators of $Y$. Thus, the errors of each model are interpreted as follows:

$$M(x) = \begin{pmatrix} Y(x) \\ \vdots \\ Y(x) \end{pmatrix} + Z(x) \text{ where } \begin{cases} \mathbb{E}[Z(x)] = 0 \\ \text{Cov}[Z(x)] = A(x) \end{cases} \tag{6}$$

Under this assumption, any linear combination with weights $\alpha_i(x)$ s.t. $\sum_{i=1}^n \alpha_i(x) = 1$ is unbiased. Thus, we define the Minimal Variance Aggregate (MVA) as the Best Linear Unbiased Estimate (BLUE) of $Y(x)$:

$$M_A(x) = \alpha_V(x)^T M(x) := \frac{\mathbb{1}^T A(x)^{-1} M(x)}{\mathbb{1}^T A(x)^{-1} \mathbb{1}} \tag{7}$$

where $\mathbb{1} := (1, .., 1)^T$ is the all ones vector. Note that $\alpha_V(x) = \text{argmin}_{\mathbb{1}^T a = 1} \mathbb{E}\left[(Y(x) - a^T M(x))^2\right]$, regardless of whether $\mathbb{E}[Z(x)] = 0$. In previous heuristics (Lee & Choi, 2014; Liang et al., 2023; Liu et al., 2016; Viana et al., 2009), $\sum_{i=1}^n \alpha_i(x) = 1$ is

assumed, while we justify it as the set of unbiased linear aggregations. This leads to a natural generalization for biased models in appendix F, while in practice bias correction did not improve results. The following section proves that despite MVA being inferior to MEA in MSE, MVA can be more robustly estimated, making MEVA better performing than MEEA.

## 3.2 A theorem on the superiority of MEVA over MEEA

To investigate the theoretical properties of MEVA and MEEA, we fix $x$ and focus on estimating $\alpha^*$ and $\alpha_V$. Details and proofs are given in appendix A. Let $Y$ be a random real-valued variable and the random vector of models $M \in \mathbb{R}^n$. Suppose we have i.i.d observations $(M^1, Y^1), \ldots, (M^N, Y^N)$, collected in the matrices $\mathcal{M} \in \mathbb{R}^{N \times n}$ and $\mathcal{Y} \in \mathbb{R}^N$. Define $M = Y\mathbb{1} + \epsilon Z$, where $Z$ is the scaled error, with $\epsilon$ chosen so that $\mathbb{E}[ZZ^T] := A$ has Frobenius norm 1 ; and $b := \mathbb{E}[YZ]$. We find that the MEA $\alpha^*$ defined in equation (3) and the MVA $\alpha_V$ (equation (7)) satisfy:

$$\alpha^* = \lambda \alpha_V + (1 - \lambda)\alpha_R \text{ where } \begin{cases} \lambda &= \frac{s(E[Y^2] - u)}{s(E[Y^2] - u) + (\epsilon + t)^2} \\ \alpha_R &= \frac{1}{\epsilon + t}A^{-1}b \end{cases} \tag{8}$$

with $s = \mathbb{1}^T A^{-1} \mathbb{1}$, $t = b^T A^{-1} \mathbb{1}$ and $u = b^T A^{-1} b$ assumed to be non zero quantities. Note that $\lambda \in [0, 1]$, making $\alpha^*$ a convex combination of $\alpha_V$ and $\alpha_R$. For the empirical variants of these aggregations, we use $\hat{A} = \frac{1}{\epsilon^2 N}(\mathcal{M}^T - \mathbb{1}\mathcal{Y}^T)(\mathcal{M} - \mathcal{Y}\mathbb{1}^T)$. Using the central limit theorem on $\hat{A} - A := \frac{1}{\sqrt{N}}\delta A$, we find that the scaled error $\delta A$ converges to a Gaussian Matrix of finite covariance. Similarly, $\mathbb{E}[Y^2]$ and $b$ are estimated with estimators converging at $\frac{1}{\sqrt{N}}$ rates, with $\delta V$ and $\delta b$ their scaled error. Write $\hat{\alpha}_E := (\mathcal{M}\mathcal{M}^T)^{-1}\mathcal{M}^T\mathcal{Y}$ for the empirical MEA, $\hat{\alpha}_V = (\hat{A}^{-1}\mathbb{1})/(\mathbb{1}^T \hat{A}^{-1}\mathbb{1})$ the empirical MVA, and $\mathcal{L}(\alpha) = \mathbb{E}[(Y - \alpha^T M)^2]$ the Mean Squared Error. We prove the following theorem:

**Theorem 1.** *Assume $s, u, t \neq 0$, then there exist two sequences of random variables, $K_N^E$ and $K_N^V$, each of which converges in distribution to a distinct finite random variable, such that*

$$\mathcal{L}(\hat{\alpha}_V) = \frac{1}{\lambda}\mathcal{L}(\alpha^*) + \frac{1}{N}K_N^V \text{ and } \mathcal{L}(\hat{\alpha}_E) = \mathcal{L}(\alpha^*) + \frac{1}{\sqrt{N}}K_N^E \tag{9}$$

MEVA incurs a constant error relative to MEEA, given by $\frac{1}{\lambda}$. However, MEVA is less sensitive to perturbations, causing its error to converge more rapidly to its limiting value than MEEA. Thus, in the model aggregation setting where $\lambda$ is close to $1$[1], the smaller difference in limiting loss values is outweighed by MEVA's faster convergence making it outperform MEEA

While this theorem applies to a fixed $x$ and $N$ observations, we are interested in the case where these $N$ data points are used to extrapolate across all $x$. In such cases, model aggregation inherently operates in the scarce data regime where $\hat{A} - A$ is large. This amplifies the behavior described in the theorem, further emphasizing the practical advantages of MEVA.

## 3.3 Computing the aggregation: the Minimal Empirical Variance Aggregate (MEVA)

To compute our empirical aggregation $\hat{\alpha}_V(x)$, we must compute the covariance matrix of the error for all $x \in \mathcal{X}$ from the data. This poses several challenges when $A(x)$ is unknown. First, we must approximate a symmetric matrix everywhere on the domain, needing $n(n + 1)/2$ coefficients to aggregate $n$ models. Moreover, we must ensure this matrix is always positive-definite. To simplify the approximation and to easily enforce the positive-definite condition, we suppose that for all $x$, $A(x)$ can be diagonalized using a fixed eigenbasis $P$. Then, its positive eigenvalues are approximated by $e^{\lambda_i(x)}$, where the $\lambda_i$ are functions to be regressed from the data. Thus, we approximate $A(x)$ with $\hat{A}(x) = P^T Diag(e^{\lambda_1(x)}, .., e^{\lambda_n(x)})P$. This formulation has the advantage of symmetrizing the covariance and precision matrix estimation, as the log-eigenvalues of both matrices are opposite. This is desirable because while we are interested in the precision matrix, the covariance matrix is easier to estimate. As a further simplifying assumption, we will take $P = I_n$, which is equivalent to assuming that models have independent errors. Appendix D presents the general case $P \neq I_n$, which

---

[1]See appendix A for more details

in our experiments did not improve performance. Using this assumption on $\hat{A}(x)$ in equation 7, we get:

$$
\begin{cases}
\hat{A}(x) & = & Diag(e^{\lambda_1(x)}, .., e^{\lambda_n(x)}) & \text{where} & \lambda_i \text{ are regressors} \\
\hat{\alpha}_V(x) & = & Softmax(-\lambda(x)) & \text{where} & Softmax(-\lambda(x))_i = \frac{e^{-\lambda_i(x)}}{\sum_{k=1}^{n} e^{-\lambda_k(x)}}
\end{cases}
\tag{10}
$$

Notice that using our model of the error (6), we have re-discovered the popular softmax, used in particular in Mixture-of-Experts (MoE). All that remains for our aggregation to be complete is to train the regressors $\lambda_i$. To do so, we will rely on covariance matrix estimation. Let $Z^1, .., Z^N$ be i.i.d. samples of the random variable $Z \in \mathbb{R}^n$ such that $\mathbb{E}[Z] = 0$. An unbiased estimate of $Cov[Z]$ is

$$
\hat{\text{Cov}}\, Z = \frac{1}{N} \sum_{i=1}^{N} Z^i (Z^i)^T = \underset{\Sigma \in \mathbb{R}^{n \times n}}{\operatorname{argmin}} \sum_{i=1}^{N} \|\Sigma - Z^i (Z^i)^T\|_F^2
\tag{11}
$$

where $\|\cdot\|_F$ is the Frobenius norm. If we further assume that $\hat{\text{Cov}}\, Z$ must be diagonal (as we do with $\hat{A}(x)$), then its coefficients are such that $(\hat{\text{Cov}}\, Z)_{kk} = \operatorname{argmin}_{c \in \mathbb{R}} \sum_{i=1}^{N} \|c - (Z_k^i)^2\|_2^2$. We will now regularize this $\operatorname{argmin}$ formulation to approximate the model errors covariance matrix for all values of $x$. Writing $e^i$ for the vector with entries defined as the sample errors $(e^i)_k = M_k(X^i) - Y(X^i)$, we identify the functions $\lambda_i$ as

$$
\lambda = \underset{l \in \mathcal{H}}{\operatorname{argmin}} \sum_{i=1}^{N} \sum_{k=1}^{n} \left( e^{l_k(X^i)} - (e^i)_k^2 \right)^2 + a\|l\|_{\mathcal{H}}^2
\tag{12}
$$

Here, we denote the entries of the vector-valued function $l$ by $l_k$, and $\mathcal{H}$ represents the space of vector-valued functions selected to approximate the log-eigenvalues. This space could be a RKHS associated with a particular kernel, a neural network with a specific architecture, or any other normed function space. The notation $\|\cdot\|_{\mathcal{H}}$ refers to the regularization norm intrinsic to that space, such as the RKHS norm or the $L_2$-norm of the weights and biases in a neural network.

# 4 EXPERIMENTS

## 4.1 SANITY CHECK: AGGREGATION ON THE BOSTON HOUSING DATASET

The Boston housing dataset (Harrison & Rubinfeld, 1978) is a popular benchmarking dataset for regression. It contains $N = 506$ samples of 14 variables. We use this dataset as a sanity check that our approach can improve upon the predictions of the aggregated models, while our focus is on the PDE examples below. We split this dataset into train, validation, and test sets, and average results over many splits. To evaluate our method, we train several standard ML methods on the train set (e.g. linear regression, gradient boosting, etc.). We then train our aggregation on the validation set before testing on the test set. To get a fair comparison, we also trained the ML methods on the train and validation sets combined so that they see the same amount of data as the aggregate. We first implement our aggregation as described in section 3.3, using a kernel method with kernel $\kappa$. This is equivalent to minimizing equation 12 over $\lambda_i \in \mathcal{H}_\kappa$ where $\mathcal{H}_\kappa$ is the RKHS defined by the kernel $\kappa$. After experimenting with many kernels, the best we found uses the Matérn kernel $\kappa_{\text{Matérn}}(u, v, \rho) = \left( 1 + \frac{\sqrt{3}\|u-v\|}{\rho} \right) \exp\left( -\frac{\sqrt{3}\|u-v\|}{\rho} \right)$, and is defined as

$$
\kappa(x, y) = \kappa_{\text{Matérn}}(M(x), M(y), \rho)
\tag{13}
$$

with adequate choice[2] of $\rho$. This formulation uses the prediction $M(x)$ to output the aggregation coefficients $\alpha(x)$.

The results of this experiment are shown in figure 2. In this situation, the mean performs well. The aggregate, denoted *MEVA with kernel*, performs better than all individual models being aggregated, their average and any ML model when fed the same data. This is a success for the MEVA, which is difficult to obtain with MEEA. The aggregation improves upon the performance of the best aggregated model, Gradient Boosting, by $r_{\text{train}} = 1 - \frac{MSE(\text{Aggregate})}{MSE(\text{Gradient Boosting})} = 7\%$, and the best model overall by $r_{\text{all}} = 2\%$.

---

[2]See details in the paper's repository

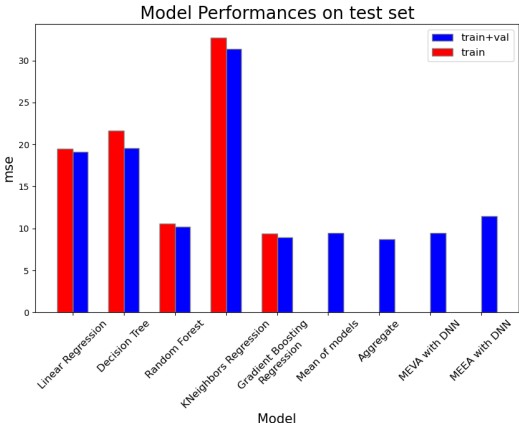

Figure 2: Comparisons of the performance of the different models on the Boston housing dataset. Red bars are the performance of models trained using the training set and used in the aggregation. Blue bars show models trained on the training and validation set (*train+val*) to get a fair comparison with the aggregations. The aggregation does not use the models trained on *train+val*.

We conduct a second experiment to demonstrate the effectiveness of our proposed method compared to Minimal Error Aggregation, defined in equation 4. In this experiment, we use a small, fully connected Deep Neural Network (DNN)[3] and train it using two different loss functions: the MEA described in section 2, and a Minimal Variance Aggregate (MVA) using an end-to-end loss especially useful for neural networks, described in appendix E. The results, presented in figure 2, highlight the impact of the chosen loss function. Since the only difference between the two approaches is the loss function, the results clearly illustrate the advantages of variance minimization. Specifically, the minimal error aggregation (denoted *MEEA with DNN* in the figure) performs significantly worse compared to our minimal variance aggregation (denoted *MEVA with DNN* in the figure). It also performs worse than the two best models aggregated.

## 4.2 AGGREGATION OF PDE SOLVERS

This section introduces the approach for combining Partial Differential Equation (PDE) solvers through model aggregation. By incorporating classical PDE solvers into the aggregation framework, we lower the accuracy threshold necessary for the aggregated solution to outperform individual models. Notably, this threshold is significantly below the typical 1% relative error benchmark targeted by most ML-based PDE solvers. We first formulate the aggregation problem in the operator learning context, followed by two experimental case studies demonstrating the effectiveness of this approach.

### 4.2.1 OPERATOR LEARNING IN THE AGGREGATION CONTEXT

Consider the Laplace equation with zero Dirichlet boundary conditions on the domain $\Omega = [0,1]^2$ as an illustrative example of a PDE:

$$\begin{cases} -\Delta u^\dagger(\omega) = f(\omega) & \text{for} \quad \omega \in \Omega \\ u^\dagger(\omega) = 0 & \text{for} \quad \omega \in \partial\Omega \end{cases} \quad (14)$$

where $f \in \mathcal{F} \subset L^2(\Omega)$ and $u^\dagger \in \mathcal{Y} \subset H^2(\Omega) \cap H_0^1(\Omega)$ is the solution. The solution operator maps the source term $f$ to the solution $u^\dagger$. In this context, the solution operator is defined as $S : f \in \mathcal{F} \mapsto u^\dagger \in \mathcal{Y}$. This operator is often approximated using a PDE solver, which can be based on methods like finite element analysis (Bathe, 2008) or spectral methods (Fornberg & Sloan, 1994). Alternatively, the operator can be learned directly from pairs of input/output solutions, a supervised learning task known as operator learning (Kovachki et al., 2024). Since PDE solvers and machine learning methods approximate operators, we can naturally frame our aggregation task as an

---

[3]See details in the paper's repository

aggregation of multiple operators (or PDE solvers), denoted $M_1, \ldots, M_n$. The aggregated operator $M_A$ is then expressed using $\alpha$, an operator that maps $\mathcal{F}$ to $L^2(\Omega, \mathbb{R}^n)$, so that:

$$M_A(f) = \sum_{i=1}^{n} \alpha_i(f) M_i(f) \in \mathcal{F} \tag{15}$$

**Remark:** In the operator context, one may consider many aggregations that are pointwise linear with respect to the models, for instance, the aggregation of Fourier coefficients, $\tilde{M}_A(f) = \mathcal{T}^{-1}\left(\sum_{i=1}^{n} \tilde{\alpha}_i(f) \mathcal{T}(M_i(f))\right)$, where $\mathcal{T}$ represents the Fourier transform.

### 4.2.2 THE OPERATOR AGGREGATION LOSS

To get the loss for our problem, we will first state the general loss in the operator learning setting and then modify it for increased performance. Since the aggregation loss (12) is defined for real-valued objectives with an abstract input $x \in \mathcal{X}$, we must specify $\mathcal{X}$ in the context of operator learning. We use $\mathcal{X} = \{(\omega, f) \in \Omega \times \mathcal{F}\}$, which means $x = (\omega, f)$ is a choice of point in space and source term. We define $Y(x) := S(f)(\omega) = u^\dagger(\omega) \in \mathbb{R}$. Given a grid $(\omega^1, \ldots \omega^{N_1}) \in \Omega$ and a set of input functions $(f^1, \ldots, f^{N_2})$, we get the $N_1 \times N_2$ datapoints $X^{i,j} := (\omega^i, f^j)$ for our aggregation. These pairs of datapoints $(X^{i,j}, Y^{i,j} = S(f^j)(\omega^i))$ can then be used in equation 12.

When solving PDEs, the focus is often on the order of magnitude of the error, i.e., its logarithm. Therefore, we may adapt the loss function (12) to account for this. Specifically, observe that if the regularization parameter $a$ is small, then functions $\lambda$ obtained from regressors that can interpolate arbitrary data (e.g., GPs with universal kernels) may result in a near-zero loss and $e^{\lambda_k(X^{i,j})} \rightarrow (Y(X^{i,j}) - M_k(X^{i,j}))^2 := e_k^{i,j}$ as $a \rightarrow 0$. Thus, we can linearize the exponential around $\log[e_k^{i,j}]$ to see that:

$$(e^{\lambda_k(X^{i,j})} - e_k^{i,j})^2 \approx \left(l_k(X^{i,j}) - \log[e_k^{i,j}]\right)^2 (e_k^{i,j})^4 \tag{16}$$

This linearization shows that samples where the models perform well are down-weighted, and the aggregation would focus on instances where they perform poorly. Instead, we use the following more sensitive loss (17) where the down-weighting is removed, which, in practice, gives good results on these problems:

$$\lambda_s = \operatorname*{argmin}_{l \in \mathcal{H}} \sum_{i=1}^{N_1} \sum_{j=1}^{N_2} \sum_{k=1}^{n} \left(l_k(X^{i,j}) - \log\left[(Y(X^{i,j}) - M_k(X^{i,j}))^2\right]\right)^2 + a\|l\|_{\mathcal{H}}^2 \tag{17}$$

We also recall the process for obtaining $\alpha$ and the aggregate $M_A$ from $\lambda_s$ is given in equations 10 and 15. In both examples, we use the Fourier Neural Operator (FNO) (Li et al., 2021) to learn the operator $l$. We also explored a kernel-based approach for operator learning, as detailed in Batlle et al. (2024). Although operator learning using a Matérn kernel yielded good results, FNO offered better performance with minimal hyperparameter tuning for the examples discussed in this section.

### 4.3 THE LAPLACE EQUATION

Our first example is the Laplace equation (14). We use six simple solvers as models: a finite difference method, two finite differences with an inhomogeneous grid (one denser for $x < 0.4$, the other for $x > 0.4$), a spectral method, and a GP solver. Our data is generated by sampling[4] random parameters $f_{max}, \mu_0, \mu_1, R$ for functions of the form

$$u(x,y) = -\sin(\pi x) \cdot \sin(\pi y) \cdot \sin\left(f_{max} \cdot \exp\left(-\begin{bmatrix} x - \mu_0 \\ y - \mu_1 \end{bmatrix}^T \cdot R \cdot \begin{bmatrix} x - \mu_0 \\ y - \mu_1 \end{bmatrix}\right)\right) \tag{18}$$

Six hundred pairs of functions $(f_i := \Delta u_i, u_i)$ are sampled, of which 500 are used for training and 100 for testing. Figure 6 shows an example of such a pair. These functions and the models are evaluated on a grid of $100 \times 100$. The aggregation is performed by FNO, which takes input $f$ and the models' outputs and outputs the aggregation coefficients' logits $\lambda$. When training a FNO with parameters $\theta \in \Theta \subset \mathbb{R}^{N_\Theta}$ and loss (17), we find the minimizer of (17) with $\mathcal{H} = \{f \mapsto FNO(\theta)(f, M_1(f), \ldots, M_n(f)) \text{ for } \theta \in \Theta\}$ with $\|FNO(\theta)\|_{\mathcal{H}} = \|\theta\|_2$.

---

[4]See details in the paper's repository

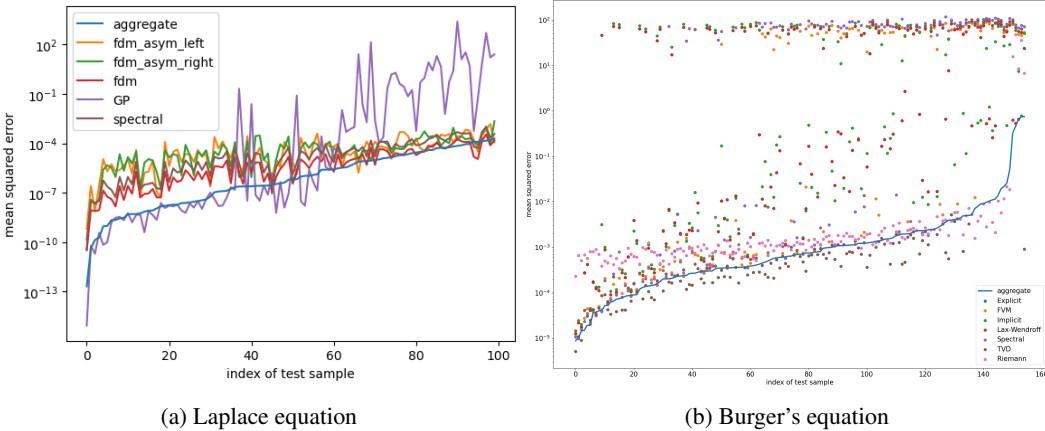

(a) Laplace equation          (b) Burger's equation

Figure 3: $\log$ MSE of the different methods for: (3a) the Laplace equation; (3b) Burger's equation. Samples are sorted by the error of the aggregate

Apart from the GP solver, all other methods perform similarly, with the finite difference method being the best overall. The GP solver, implemented with a fixed length scale, has two regimes. When the function $u_i$ is not evolving too fast (i.e. when $f_{max}$ is not too large), the kernel is well specified, and the GP solver obtains the best performance. On other occasions where $f_{max}$ is larger, the method fails and has a substantial error. Thus, the aggregation challenge here is to use the GP when performing well but ignore it when it fails. The result of this experiment is shown in figure 3a. Because of the significant error in the failure case, we observe that the mean is a poor aggregation that performs worse than all models aggregated. Our aggregate is consistently among the best performers on each sample and the best-performing model for many samples. On average, our aggregate performs one order of magnitude better than the models aggregated, as shown in table 1. An example of model outputs, errors, $\alpha$, and final aggregation is shown in figure 7. As a comparison, we also train FNO using a direct MSE loss of the aggregation, akin to (4). We observe that this method fails as the aggregation converges to a constant, i.e., $\alpha(f, \omega) \approx (1, 0, 0, 0, 0, 0) \; \forall \; f, \omega$, picking the best-performing model. We believe that this convergence to a fixed value, without combining the different models to get a better one as observed in our method, is due to the absence of the $\log$ term that we added in the sharp loss (17). As we observed in section 4.2.2, without this $\log$ it is difficult for a loss to differentiate between $10^{-3}$ and $10^{-9}$ error. Furthermore, while learning the log-error instead of the error is a simple modification from (12) to (17), it is difficult to see where this $\log$ would be added in a loss like (4).

## 4.4 BURGER'S EQUATION

We now turn to a non-linear PDE, Burger's equation with periodic boundary condition and finite time interval:

$$\begin{cases} \partial_t u(t,x) + \partial_x(\frac{1}{2}u^2(t,x)) = & \nu \partial_{xx} u(t,x) & \text{for} & x, t \in [0,1]^2 \\ u(0,x) = f(x) & \text{for} & x \in [0,1] \\ u(t,0) = u(t,1) & \text{for} & t \in [0,1] \end{cases} \quad (19)$$

We choose $\nu = 2.10^{-3}$ and pick initial conditions $f_i$ as samples from the Gaussian process $\mathcal{N}(0, K_{\exp \sin^2})$ where $K_{\exp \sin^2}(x,y) = \exp\left(-\frac{2\sin^2(\pi|x-y|)}{l^2}\right)$ and $l = 1.5$. With this set of parameters, the initial conditions are infinitely differentiable periodic functions that often, but not always, form a shock within the time frame studied ($t \in [0,1]$). We use seven different solvers with the same fixed discretization grid. While the first five methods implement the viscid version of Burger's equation directly, for instance, using an explicit scheme or a finite volume method, the penultimate uses a flux limiter. The last one solves the inviscid Burger's equation. With this variety of solvers, we have methods that are accurate in smooth cases when there is no large shock, an intermediate method that may be more robust to shocks, and a method that will never diverge even in the presence of shocks, but will be less accurate in general because it solves a slightly different equation. A good

aggregation must be able to recognize artefacts and divergences that may arise around shocks but use the more precise, although more brittle, solvers in cases where they are accurate. To perform the aggregation, similarly to the previous section, we use FNO, to which we feed the models' prediction as a stack of 2-dimensional arrays (one for time and one for space). Specifically, we minimize loss (17) with $\lambda(f) = FNO(\theta)(M_1(f), .., M_n(f))$ for some $\theta$.

The results are displayed in figure 3b. Our method successfully leverages the predictions of the different models to give a robust and accurate prediction. The behaviors of our aggregation can be separated into a few cases. On the left of the graph, we find easy initial conditions that all models can approximate correctly. We may notice the Riemann method, which solves the inviscid Burger's equation, has lower accuracy. Thus, on this easy half of the graph, the aggregation mainly relies on the TVD solver, the most precise, to achieve excellent accuracy. In the right half, we find the more complex cases where the aggregation has an accuracy closer to the more robust Riemann solver. In this half, there are cases where the TVD method fails and diverges and is avoided by our aggregation. The process can identify each solver's strengths and weaknesses and obtain a better average error, as shown in table 2. An example of model outputs, errors, $\alpha$, and final aggregation is shown in figure 8.

| Method | Geometric Mean of MSE ($\log_{10}$) |
|---|---|
| **Aggregate** | **-6.282** |
| FDM | -5.523 |
| Spectral | -4.988 |
| GP | -4.739 |
| FDM (Asymmetric Right) | -4.685 |
| FDM (Asymmetric Left) | -4.699 |

Table 1: Result of Laplace equation experiment (sec. 4.3)

| Method | Geometric Mean of MSE ($\log_{10}$) |
|---|---|
| **Aggregate** | **-3.106** |
| Riemann | -2.734 |
| TVD | -2.568 |
| FVM | -1.228 |
| Spectral | -0.625 |
| Implicit | -0.488 |
| Explicit | -0.455 |
| Lax-Wendroff | -0.455 |

Table 2: Result of Burger's equation experiment (sec. 4.4)

## 5 DISCUSSION

We introduced a general data-driven framework for aggregating models with minimal assumptions. As demonstrated in the pathological example Section 2.2 and the theorem 3.2, directly training the aggregate model is unstable and may result in an aggregator that fails to improve upon the performance of individual models. To address this, we introduced a simple assumption of unbiased models in (6) and reformulated the aggregation task as a variance minimization problem (Section 3). In one data science problem (Section 4.1) and two PDE-solving examples (Sections 4.3 and 4.4), our loss function consistently outperformed the minimal error aggregate and led to an aggregation that surpassed the performance of the individual models. These latter two examples also illustrate the flexibility of our method, which can aggregate not only machine learning models but also other types of models, such as PDE solvers, in a data-driven manner. Notably, we used FNO to get an aggregate performance much higher than it could get when trying to solve the PDE directly.

Our method relies on the availability of unseen data to estimate the error of different models. Therefore, as seen in the Boston Housing example (section 4.1), splitting the data further to create an unseen validation set in scenarios where data is limited may significantly reduce the amount of training data available for the models, potentially impairing overall performance. Additionally, in challenging aggregation tasks, there is no guarantee that the aggregate will consistently outperform the individual models or simpler methods, such as averaging, despite the higher computational cost.

MEVA's flexibility makes it well-suited for integrating diverse methods, particularly in structured contexts like scientific computing, where non-intrusive solutions are often necessary. By requiring minimal assumptions about the models or data, MEVA can effectively aggregate legacy solvers that are difficult to modify, physics-based models built on differing assumptions, or models with incompatible software frameworks.

ACKNOWLEDGEMENTS

TB and HO acknowledge support from the Air Force Office of Scientific Research under MURI awards number FA9550-20-1-0358 (Machine Learning and Physics-Based Modeling and Simulation), FOA-AFRL-AFOSR-2023-0004 (Mathematics of Digital Twins) and by the Department of Energy under award number DE-SC0023163 (SEA-CROGS: Scalable, Efficient and Accelerated Causal Reasoning Operators, Graphs and Spikes for Earth and Embedded Systems). Additionally HO acknowledge support from the DoD Vannevar Bush Faculty Fellowship Program.

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

## A   THEOREM ON THE STABILITY OF THE MEVA

We consider the setting without a dependence on $x$ to prove a theorem on the robustness of MEVA compared to MEEA. Let $Y$ be a random real-valued variable and the random vector of models $M \in \mathbb{R}^n$. Suppose we have i.i.d observations $(M^1, Y^1), \ldots, (M^N, Y^N)$, collected in the matrices $\mathcal{M} \in \mathbb{R}^{N \times n}$ and $\mathcal{Y} \in \mathbb{R}^N$. We wish to study the Mean Squared Error (MSE) of an aggregation method of the form $M_A = \alpha^T M$. Define

$$\mathcal{L}(\alpha) = \mathbb{E}\left[(Y - \alpha^T M)^2\right] \tag{20}$$

Let $C = \mathbb{E}\left[MM^T\right]$, $\gamma = \mathbb{E}\left[YM\right]$, we have that the loss is minimized at $\alpha^* = C^{-1}\gamma$ the MEA. Let $M = Y\mathbb{1} + \epsilon Z$, where $Z$ is the scaled error, with $\epsilon$ chosen so that $\mathbb{E}\left[ZZ^T\right] := A$ has Frobenius norm 1 ; and $b := \mathbb{E}\left[YZ\right]$. We have that

$$\gamma = E[Y^2]\mathbb{1} + \epsilon b \tag{21}$$
$$C = \epsilon^2 A + (E[Y^2]\mathbb{1} + \epsilon b)1^T + \epsilon 1 b^T \tag{22}$$

### A.1   CHARACTERIZATION OF $\alpha^*$

To facilitate our computations, we introduce $D = \epsilon^2 A + (E[Y^2]\mathbb{1} + \epsilon b)1^T$. We define

$$s = \mathbb{1}^T A^{-1}\mathbb{1}, \tag{23}$$
$$t = b^T A^{-1}\mathbb{1}. \tag{24}$$
$$u = b^T A^{-1}b. \tag{25}$$
$$v = \mathbb{1}^T A^{-1}\left(E[Y^2]\mathbb{1} + \epsilon b\right) = E[Y^2]s + \epsilon t \tag{26}$$

Using the Sherman & Morrison (1950) formula, we find that

$$D^{-1} = \frac{1}{\epsilon^2}\left(A^{-1} - \frac{1}{\epsilon^2 + v}A^{-1}(E[Y^2]1 + \epsilon b)1^T A^{-1}\right) \tag{27}$$
$$C^{-1} = D^{-1} - \frac{\epsilon}{1+w}D^{-1}\mathbb{1}b^T D^{-1}, \text{ where } w = \epsilon b^t D^{-1}\mathbb{1} \tag{28}$$

Let $\alpha, \beta, \delta, \zeta$ such that

$$D^{-1}\mathbb{1} = \alpha A^{-1}\mathbb{1} + \beta A^{-1}b \tag{29}$$
$$D^{-1}b = \delta A^{-1}\mathbb{1} + \zeta A^{-1}b \tag{30}$$

Thus, $w = \epsilon b^t D^{-1}1 = \epsilon(\alpha t + \beta u)$ and $b^T D^{-1}b = \delta t + \zeta u$. The coefficients are:

$$\begin{aligned}
\alpha &= \frac{1}{\epsilon^2} \cdot \frac{\epsilon^2 + \epsilon t}{\epsilon^2 + v}, \quad \beta = -\frac{1}{\epsilon} \cdot \frac{s}{\epsilon^2 + v}, \\
\delta &= \frac{-E[Y^2]t}{\epsilon^2(\epsilon^2 + v)}, \quad \zeta = \frac{\epsilon^2 + E[Y^2]s}{(\epsilon^2 + v)\epsilon^2}.
\end{aligned} \tag{31}$$

The computation of $C^{-1}\gamma$ is broken into two parts:

$$C^{-1}\gamma = C^{-1}\left(E[Y^2]\mathbb{1} + \epsilon b\right) = E[Y^2]C^{-1}\mathbb{1} + \epsilon C^{-1}b \tag{32}$$

The intermediate terms are:

$$C^{-1}\mathbb{1} = \frac{1}{1+w}D^{-1}\mathbb{1}, \tag{33}$$
$$C^{-1}b = D^{-1}b - \frac{\epsilon(b^T D^{-1}b)}{1+w}D^{-1}\mathbb{1}. \tag{34}$$

Substitute $D^{-1}\mathbb{1}$ and $D^{-1}b$:

$$C^{-1}\mathbb{1} = \frac{\alpha}{1+w}A^{-1}\mathbb{1} + \frac{\beta}{1+w}A^{-1}b \tag{35}$$

$$C^{-1}b = \left(\delta - \frac{\epsilon(\delta t + \zeta u)}{1+w}\alpha\right)A^{-1}\mathbb{1} + \left(\zeta - \frac{\epsilon(\delta t + \zeta u)}{1+w}\beta\right)A^{-1}b. \tag{36}$$

Write $C^{-1}\gamma = xA^{-1}\mathbb{1} + yA^{-1}b$, then

$$x = E[Y^2]\frac{\alpha}{1+w} + \epsilon\delta - \frac{\epsilon^2(\delta t + \zeta u)}{1+w}\alpha \tag{37}$$

$$y = E[Y^2]\frac{\beta}{1+w} + \epsilon\zeta - \frac{\epsilon^2(\delta t + \zeta u)}{1+w}\beta. \tag{38}$$

Using Sympy (Meurer et al., 2017) to simplify these expressions, we find that:

$$x = \frac{E[Y^2] - u}{s(E[Y^2] - u) + (\epsilon + t)^2} \tag{39}$$

$$y = \frac{t + \epsilon}{s(E[Y^2] - u) + (\epsilon + t)^2} \tag{40}$$

Defining $\alpha_V = \frac{A^{-1}\mathbb{1}}{\mathbb{1}^T A^{-1}\mathbb{1}}$ and $\alpha_R = \frac{A^{-1}b}{\epsilon + \mathbb{1}^T A^{-1}b}$, we have that

$$\alpha^* = \lambda\alpha_V + (1 - \lambda)\alpha_R \text{ where } \lambda = \frac{s(\mathbb{E}[Y^2] - u)}{s(\mathbb{E}[Y^2] - u) + (\epsilon + t)^2} \tag{41}$$

One may find that

$$\mathbb{E}[Y^2] - u = \mathbb{E}[Y]^2 -$$
$$\mathbb{E}[Y(M - Y\mathbb{1}))]^T \mathbb{E}[(M - Y\mathbb{1}))(M - Y\mathbb{1}))^T]^{-1}\mathbb{E}[Y(M - Y\mathbb{1}))] \geq 0$$

using the Schur complement of the covariance matrix of the vector $(Y, M^T - Y\mathbb{1}^T)^T \in \mathbb{R}^{n+1}$. Thus, $\lambda \in [0, 1]$, and $\alpha^*$ is a convex combination of the MVA, $\alpha_V$, and $\alpha_R$. Note that

$$\mathbb{1}^T C^{-1}\gamma = x * s + y * t = 1 - \epsilon y \tag{42}$$

and

$$\gamma^T C^{-1}\gamma = \mathbb{E}[Y^2](y * t + x * s) + \epsilon * (y * u + x * t) \tag{43}$$

$$= \frac{(E[Y^2])^2 s + 2E[Y^2]\epsilon t - E[Y^2]su + E[Y^2]t^2 + \epsilon^2 u}{s(E[Y^2] - u) + (\epsilon + t)^2} \tag{44}$$

$$= \mathbb{E}[Y^2] - \epsilon^2 \frac{\mathbb{E}[Y^2] - u}{s(E[Y^2] - u) + (\epsilon + t)^2} \tag{45}$$

Thus, we have that

$$\mathcal{L}(\alpha^*) = \frac{\lambda\epsilon^2}{s} = \lambda\mathcal{L}(\alpha_V) \tag{46}$$

and

$$\mathcal{L}(\alpha_V) = \epsilon^2\alpha_V^T A\alpha_V = \frac{\epsilon^2}{s} \tag{47}$$

## A.2 PERTUBATION OF $\alpha_V$

We want to study the finite sample properties of the empirical estimates, thus we define

$$\hat{A} = \frac{1}{\epsilon^2 N}(\mathcal{M}^T - \mathbb{1}\mathcal{Y}^T)(\mathcal{M} - \mathcal{Y}\mathbb{1}^T) := A + \frac{1}{\sqrt{N}}\delta A \tag{48}$$

$$\hat{b} = \frac{1}{\epsilon N}(\mathcal{M}^T - \mathbb{1}\mathcal{Y}^T)\mathcal{Y} := b + \frac{1}{\sqrt{N}}\delta b \tag{49}$$

$$\widehat{\mathbb{E}[Y^2]} = \frac{1}{N}\|\mathcal{Y}\|_2^2 := \mathbb{E}[Y^2] + \frac{1}{\sqrt{N}}\delta V \tag{50}$$

The central limit theorem asserts that $\delta A$, $\delta b$ and $\delta V$ converge in distribution to a Gaussian random variable, with finite covariance matrices. If we study the perturbation of $\alpha_V$ when estimating it using $\hat{A}$, we have that $\mathcal{L}(\alpha_V) = \epsilon^2 \frac{\mathbb{1}\hat{A}^{-1}A\hat{A}^{-1}\mathbb{1}}{(\mathbb{1}\hat{A}^{-1}\mathbb{1})^2}$. At the first order in $\frac{1}{\sqrt{N}}$,

$$\hat{A}^{-1}\mathbb{1} = A^{-1}\mathbb{1} - \frac{1}{\sqrt{N}}A^{-1}\delta A A^{-1}\mathbb{1} \tag{51}$$

$$\mathbb{1}^T\hat{A}^{-1}\mathbb{1}^2 = \mathbb{1}^TA^{-1}\mathbb{1}^2 - 2\frac{1}{\sqrt{N}}\mathbb{1}^TA^{-1}\mathbb{1}\mathbb{1}^TA^{-1}\delta A A^{-1}\mathbb{1} \tag{52}$$

$$\mathbb{1}^T\hat{A}^{-1}A\hat{A}^{-1}\mathbb{1} = \mathbb{1}^TA^{-1}\mathbb{1} - 2\frac{1}{\sqrt{N}}\mathbb{1}^TA^{-1}\delta A A^{-1}\mathbb{1} \tag{53}$$

$$\tag{54}$$

Using the first order expansion of a fraction,

$$\frac{M + \eta N}{O + \eta Q} = \frac{M}{O} + \eta\left(\frac{N}{O} - \frac{MQ}{O^2}\right) + \mathcal{O}(\eta^2) \tag{55}$$

We find that first-order terms cancel out, meaning that

$$\mathcal{L}(\hat{\alpha}_V) = \mathcal{L}(\alpha_V) + \mathcal{O}(\frac{\epsilon^2}{N}) \tag{56}$$

$\mathcal{O}(\frac{\epsilon^2}{N}) = \frac{\epsilon^2}{N}f(\delta A)$, where $f$ is a continuous function of $\delta A$ such that $f(\delta A)/\|\delta A\|^2 = \mathcal{O}(1)$ as $\delta A \to 0$. Since $\delta A$ converges in distribution to a Gaussian $Z$ with finite covariance, using the continuous mapping theorem Billingsley (1999), we have that $f(\delta A)$ converges to $f(Z)$ in distribution, a finite random variable.

### A.3 PERTURBATION OF $\alpha^*$

#### A.3.1 PERTURBATIONS OF $t$, $u$, AND $s$

If $(A + \frac{1}{\sqrt{N}}\delta A)(x + \delta x) = y + \frac{1}{\sqrt{N}}\delta y$, then at the first order $\delta x = \frac{1}{\sqrt{N}}A^{-1}(\delta y - \delta A x)$:

$$\delta s = -\mathbb{1}^TA^{-1}(\delta A)A^{-1}\mathbb{1},$$
$$\delta t = \mathbb{1}^TA^{-1}\delta b - \mathbb{1}^TA^{-1}(\delta A)A^{-1}b,$$
$$\delta u = 2b^TA^{-1}\delta b - b^TA^{-1}(\delta A)A^{-1}b.$$

#### A.3.2 PERTURBATION OF THE DENOMINATOR

The denominator of $C^{-1}\gamma$ is:

$$D = s(E[Y^2] - u) + (\epsilon + t)^2.$$

The perturbation is:

$$\sqrt{N}(\hat{D} - D) = \delta D = \delta s(E[Y^2] - u) + s\delta v - s\delta u + 2(\epsilon + t)\delta t.$$

#### A.3.3 PERTURBATION OF THE NUMERATOR

The numerator of $\hat{C}^{-1}\hat{\gamma}$ is:

$$\hat{\mathcal{N}} = (E[\hat{Y}^2] - \hat{u})\hat{A}^{-1}\mathbb{1} + (\epsilon + \hat{t})\hat{A}^{-1}\hat{b}.$$

#### A.3.4 FIRST TERM: $(E[\hat{Y}^2] - \hat{u})\hat{A}^{-1}\mathbb{1}$

Using $E[\hat{Y}^2] = E[Y^2] + \delta v$, $\hat{u} = u + \delta u$, and the perturbation formula for $\hat{A}^{-1}\mathbb{1}$:

$$(E[\hat{Y}^2] - \hat{u})\hat{A}^{-1}\mathbb{1} = (E[Y^2] - u)A^{-1}\mathbb{1} - \frac{1}{\sqrt{N}}(E[Y^2] - u)A^{-1}(\delta A)A^{-1}\mathbb{1} + \frac{1}{\sqrt{N}}(\delta v - \delta u)A^{-1}\mathbb{1}.$$

### A.3.5 SECOND TERM: $(\epsilon + \hat{t})\hat{A}^{-1}\hat{b}$

Using $\hat{t} = t + \delta t$, $\hat{b} = b + \delta b$, and the perturbation formula for $\hat{A}^{-1}\hat{b}$:

$$(\epsilon + \hat{t})\hat{A}^{-1}\hat{b} = (\epsilon + t)A^{-1}b + \frac{1}{\sqrt{N}}\left[\delta t A^{-1}b + (\epsilon + t)A^{-1}\delta b - (\epsilon + t)A^{-1}(\delta A)A^{-1}b.\right]$$

### A.3.6 FINAL EXPRESSION FOR $\hat{\mathcal{N}}$

Combining the two terms:

$$\hat{\mathcal{N}} = (E[Y^2] - u)A^{-1}\mathbb{1} + (\epsilon + t)A^{-1}b + \frac{1}{\sqrt{N}}\left[-(E[Y^2] - u)A^{-1}(\delta A)A^{-1}\mathbb{1} + (\delta v - \delta u)A^{-1}\mathbb{1}\right. \tag{57}$$

$$\left. + \delta t A^{-1}b + (\epsilon + t)A^{-1}\delta b - (\epsilon + t)A^{-1}(\delta A)A^{-1}b.\right] \tag{58}$$

### A.3.7 FULL PERTURBATION OF $C^{-1}\gamma$

The final perturbed $C^{-1}\gamma$ is:

$$\hat{C}^{-1}\hat{\gamma} = \frac{\hat{\mathcal{N}}}{\hat{D}}.$$

Where:

$$\hat{D} = D + \frac{1}{\sqrt{N}}\delta D, \quad \hat{\mathcal{N}} = \mathcal{N} + \frac{1}{\sqrt{N}}\delta\mathcal{N},$$

and:

$$\mathcal{N} = (E[Y^2] - u)A^{-1}\mathbb{1} + (\epsilon + t)A^{-1}b.$$

Now, we see that $\hat{\mathcal{N}}^T A\hat{\mathcal{N}} = \mathcal{N}^T A\mathcal{N} + 2\mathcal{N}^T A\delta\mathcal{N}$ at the first order. We have that

$$\sqrt{N}A\delta\mathcal{N} = -(E[Y^2] - u)\delta A A^{-1}\mathbb{1} + (\delta v - \delta u)\mathbb{1} + \delta t b \tag{59}$$

$$+ (\epsilon + t)\delta b - (\epsilon + t)\delta A A^{-1}b \tag{60}$$

$$\sqrt{N}\mathcal{N}A\delta\mathcal{N} = (E[Y^2] - u)\left[-(E[Y^2] - u)\mathbb{1}^T A^{-1}\delta A A^{-1}\mathbb{1} + (\delta v - \delta u)s + t\delta t\right. \tag{61}$$

$$+ (\epsilon + t)(\mathbb{1}^T A^{-1}\delta b - \mathbb{1}^T A^{-1}\delta A A^{-1}b] \tag{62}$$

$$+ (\epsilon + t)\left[-(E[Y^2] - u)b^T A^{-1}\delta A A^{-1}\mathbb{1} + (\delta v - \delta u)t\right. \tag{63}$$

$$+ u\delta t + (\epsilon + t)(b^T A^{-1}\delta b - b^T A^{-1}\delta A A^{-1}b] \tag{64}$$

Then, we have that

$$\sqrt{N}b^t\delta\mathcal{N} = -(E[Y^2] - u)b^T A^{-1}\delta A A^{-1}\mathbb{1} + (\delta v - \delta u)t \tag{65}$$

$$+ u\delta t + (\epsilon + t)(b^T A^{-1}\delta b - b^T A^{-1}\delta A A^{-1}b) \tag{66}$$

We have:

$$\hat{y}b^T\hat{\alpha} = \frac{1}{\hat{D}}(\epsilon + \hat{t})b^T\hat{\alpha} = \frac{1}{\hat{D}}\left[(\epsilon + t)b^T\alpha + \frac{1}{\sqrt{N}}\delta t b^t\alpha + \frac{1}{\sqrt{N}}(\epsilon + t)b^T\frac{\delta\mathcal{N}}{\hat{D}}\right] \tag{67}$$

and $\hat{y}^2 = \frac{(\epsilon + t)^2 + 2\frac{\epsilon}{\sqrt{N}}\delta t}{\hat{D}^2}$. Thus,

$$\mathcal{L}(\hat{\alpha}_E) = \frac{\epsilon^2}{\hat{D}^2}(\mathbb{E}[Y^2]\left[(\epsilon + t)^2 + 2\epsilon\delta t\right] \tag{68}$$

$$- 2\left[(\epsilon + t)b^T\alpha\hat{D} + \delta t b^t\alpha\hat{D} + (\epsilon + t)b^T\delta\mathcal{N}\right] \tag{69}$$

$$+ \mathcal{N}^T A\mathcal{N} + 2\mathcal{N}^T A\delta\mathcal{N}) \tag{70}$$

We define

$$\mathcal{L}_o = \mathbb{E}[Y^2](\epsilon + t)^2 - 2(\epsilon + t)b^T\alpha D + \mathcal{N}^T A\mathcal{N} \tag{71}$$

$$\mathcal{L}_1 = 2\epsilon\delta t\,\mathbb{E}[Y^2] - 2\left[\delta t b^T\alpha D + (\epsilon + t)b^T\delta\mathcal{N}\right] + 2\mathcal{N}^T A\delta\mathcal{N} \tag{72}$$

We have that

$$\mathcal{L}(\hat{\alpha}_E) = \epsilon^2 \left[ \frac{\mathcal{L}_o}{D^2} + \frac{1}{\sqrt{N}} \left( \frac{\mathcal{L}_1}{D^2} - \frac{2\mathcal{L}_0 \delta D}{D^3} \right) \right] + \mathcal{O}(\frac{\epsilon^2}{N}) \tag{73}$$

Sympy computations show that

$$\mathcal{L}(\hat{\alpha}_E) = \mathcal{L}(\alpha^*) + \frac{\epsilon^2 2E[Y^2]t}{\sqrt{N}D^2} \left( -\mathbb{1}^T A^{-1} \delta b + b^T A^{-1} \delta A A^{-1} \mathbb{1} \right) + \mathcal{O}(\frac{\epsilon^2}{N}) \tag{74}$$

Here again, $\delta A$ and $\delta b$ jointly converge to Gaussian random variables, so using the continuous mapping theorem, $\xi_N = -\mathbb{1}^T A^{-1} \delta b + b^T A^{-1} \delta A A^{-1} \mathbb{1}$ converges to a Gaussian $\xi$ with mean 0 and finite covariance (because it is a linear combination of $\delta A$ and $\delta b$, which are jointly Gaussian in the limit).

## A.4 COMPARISON BETWEEN MEVA AND MEEA

Building upon the previous sections, we derived the following results for the loss functions of the two methods:

$$\mathcal{L}(\hat{\alpha}_V) = \frac{1}{\lambda} \mathcal{L}(\alpha^*) + \mathcal{O}(\frac{\epsilon^2}{N}) \tag{75}$$

$$\mathcal{L}(\hat{\alpha}_E) = \mathcal{L}(\alpha^*) + \frac{\epsilon^2 2E[Y^2]t}{\sqrt{N}D^2} \xi_N + \mathcal{O}(\frac{\epsilon^2}{N}) \tag{76}$$

$$\text{where } \lambda = \frac{s(\mathbb{E}\left[Y^2\right] - u)}{s(\mathbb{E}\left[Y^2\right] - u) + (\epsilon + t)^2} \qquad \mathcal{L}(\alpha^*) = \frac{\lambda \epsilon^2}{s} \tag{77}$$

We can reformulate these results as:

$$\mathcal{L}(\hat{\alpha}_V) = \mathcal{L}(\alpha^*) \left( 1 + \frac{(\epsilon + t)^2}{s(\mathbb{E}\left[Y^2\right] - u)} + \mathcal{O}(\frac{1}{N}) \right) \tag{78}$$

$$\mathcal{L}(\hat{\alpha}_E) = \mathcal{L}(\alpha^*) \left( 1 + \frac{2E[Y^2]t}{D(\mathbb{E}\left[Y^2\right] - u)} \frac{\xi_N}{\sqrt{N}} + \mathcal{O}(\frac{1}{N}) \right) \tag{79}$$

$$\tag{80}$$

In the context of model aggregation, we expect errors to be small, leading to $\epsilon$ being close to 0. While the MVA does not assume that $b = \mathbb{E}\left[Y \frac{M - Y \mathbb{1}}{\epsilon}\right] = 0$, a small $b$ makes $\lambda$ closer to 1. This corresponds to the errors being uncorrelated with the target. This setting of fixed $t \neq 0$ is the one studied in the theorem. We may see that even for $\lambda = 0.5$, and small $N$, say $N = 100$, the difference between $\mathcal{L}(\alpha_V)$ and $\mathcal{L}(\alpha^*)$ is a factor 2, while the difference between $\frac{1}{N}$ and $\frac{1}{\sqrt{N}}$ is a factor 10, making the latter effect much dominant.

However, $t = 0$ makes the $N^{-\frac{1}{2}}$ term disappear, resulting in MEEA and MEVA having the same convergence rate. In this scenario: $\mathcal{L}(\hat{\alpha}_V) - \mathcal{L}(\alpha^*) = \mathcal{O}(\epsilon^4 + \frac{\epsilon^2}{N})$. Thus, when t = 0, both methods perform well, with the performance gap being of the order $\epsilon^4$.
We now consider the more general case of a possibly small error correlation with the target, which leads to $t = \kappa \epsilon$. In this case,

$$\mathcal{L}(\hat{\alpha}_V) = \mathcal{L}(\alpha^*) \left( 1 + \epsilon^2 \frac{(1 + \kappa)^2}{s(\mathbb{E}\left[Y^2\right] - u)} + \mathcal{O}(\frac{1}{N}) \right) \tag{81}$$

$$\mathcal{L}(\hat{\alpha}_E) = \mathcal{L}(\alpha^*) \left( 1 + \frac{\epsilon}{\sqrt{N}} \frac{2E[Y^2]\kappa \xi_N}{D(\mathbb{E}\left[Y^2\right] - u)} + \mathcal{O}(\frac{1}{N}) \right) \tag{82}$$

$$\tag{83}$$

This comparison reduces to evaluating the relative magnitudes of $\frac{\epsilon}{\sqrt{N}}$ and $\epsilon^2$. While it is not possible to compare them a priori, our theorem is aimed at explaining the behavior of the aggregation when extrapolating to all $x$, the regime of very scarce data. For models that are accurate, it is expected that the model error $\epsilon$ is much less than the Monte-Carlo rate of the validation samples, $\frac{1}{\sqrt{N}}$, especially for $N$ not very large. In this case, MEVA remains superior to MEEA, particularly for highly accurate models.

# B   AGGREGATING PDE SOLVERS IN THE GAUSSIAN SETTING

This section describes the aggregation performance one may get with GP-based PDE solvers introduced in Chen et al. (2021). Subsection B.1 summarizes the method and subsection B.2 describes the performance of the aggregation of low-fidelity models obtained by constraining the GPs to satisfy the PDE at a small number of collocation points.

## B.1   SOLVING PDES WITH GPS AND THEIR AGGREGATION

We now recall the GP-based PDE solver introduced in Chen et al. (2021) and describe the proposed method for aggregating such solvers. Consider the Laplace equation as a prototypical example:

$$\begin{cases} -\Delta u^{\dagger}(x) = f(x) & \text{for} \quad x \in [-1,1]^2 \\ u^{\dagger}(x) = g(x) & \text{for} \quad x \in \partial([-1,1]^2) \end{cases} \tag{84}$$

Place a Gaussian prior on the solution, i.e., assume $u^{\dagger} = \xi \sim \mathcal{N}(0,\kappa)$ where $K$ is a Radial Basis Function (RBF) kernel.

Given $N$ interior collocation points $X_i \in [-1,1]^2$ and $N_b$ boundary collocation points $X_j^b \in \partial([-1,1]^2)$, we approximate the solution $u$ with the Gaussian process estimator

$$\hat{u} = \mathbb{E}\left[\xi \,\middle|\, -\Delta\xi(X) = f(X),\, \xi(X^b) = g(X^b)\right] \tag{85}$$

To express $\hat{u}$, we extend the action of $\kappa$ to distributions (linear functionals) as described in Owhadi & Scovel (2019): for two distributions $d_1, d_2$, we write

$$K(d_1, d_2) := \int d_1(x)\kappa(x,y)d_2(y)\, dx\, dy \tag{86}$$

Note that $K(\delta_x, \delta_y) = \kappa(x,y)$, while $K(\delta_x \circ \Delta, \delta_y) = \Delta_1 \kappa(x,y)$ where $\Delta$ is the laplacian, and $\Delta_1$ is the laplacian applied to the first variable.

Using this extended kernel, the Gaussian prior $\xi \sim \mathcal{N}(0,\kappa)$ can be interpreted in terms of distributions, allowing us to define $\xi(x) = \delta_x(\xi)$ and $\Delta\xi(x) = \delta_x \circ \Delta(\xi)$. In particular, this Gaussian prior implies that $\mathbb{E}[d_1(\xi)d_2(\xi)] = K(d_1, d_2)$. Defining the stack of distributions $\phi = (\delta_{X_1} \circ \Delta, .., \delta_{X_N} \circ \Delta, \delta_{X_1^b}, ..\delta_{X_{N_b}^b})$, we can define the kernel matrix

$$K(\phi, \phi)_{ij} = K(\phi_i, \phi_j) \tag{87}$$

and similarly, $K(x, \phi)_i = K(\delta_x, \phi_j)$. Using this, we have

$$\mathbb{E}\left[\xi \,\middle|\, -\Delta\xi(X),\, \xi(X^b)\right] = K(x,\phi)K(\phi,\phi)^{-1}\begin{pmatrix} -\Delta\xi(X) \\ \xi(X^b) \end{pmatrix} \tag{88}$$

In scenarios with multiple GP approximations of $u^{\dagger}$, leveraging the GP prior allows computation of correlations. Let

$$\hat{u}^{(k)} := \mathbb{E}\left[\xi \,\middle|\, -\Delta\xi(X^{(k)}),\, \xi(X^{b,k})\right] \tag{89}$$

be the distinct approximations of the solution of the PDE obtained from different interior and boundary collocation points $X^{(k)}$ and $X^{b,k}$. Write $\phi_k := (\delta_{X_1^k} \circ \Delta, .., \delta_{X_N^k} \circ \Delta, \delta_{X_1^{b,k}}, ..\delta_{X_{N_b}^{b,k}})$. Since these approximants $\hat{u}^{(k)}$, interpreted as models, are also Gaussian processes, we can explicitly compute their correlations:

$$\mathbb{E}\left[\hat{u}^{(k)}(x)\hat{u}^{(l)}(x)\right] = K(x,\phi_k)K(\phi_k,\phi_k)^{-1}\mathbb{E}\left[\begin{pmatrix} -\Delta\xi(X^k) \\ \xi(X^{b,k}) \end{pmatrix}\begin{pmatrix} -\Delta\xi(X^l) \\ \xi(X^{b,l}) \end{pmatrix}^T\right]K(\phi_l,\phi_l)^{-1}K(\phi_l,x)$$

$$= K(x,\phi_k)K(\phi_k,\phi_k)^{-1}K(\phi_k,\phi_l)K(\phi_l,\phi_l)^{-1}K(\phi_l,x)$$

Similarly, $\mathbb{E}\left[\xi(x)\hat{u}^{(k)}(x)\right] = K(x,\phi_k)K(\phi_k,\phi_k)^{-1}K(\phi_k,x)$. We then use these correlations in equation 3 to compute the aggregation analyzed in the following subsection.

### B.2 PERFORMANCE OF THE AGGREGATION

Using the GP-based PDE-solver introduced in Chen et al. (2021) and described above, we generate 100 distinct low-fidelity models of the solution of the PDE equation 84, an example of which is given in figure 4b. As this method is GP-based, the correlations outlined in equation (3) are known, and this approach reduces to nested kriging (Rullière et al., 2017). In this context, equation 3 is not only the best (in mean squared error) linear aggregate. It is also the best non-linear aggregate of all models. Figure 4 shows the results of this experiment. While the individual models are far from approximating the solution due to the limited number of collocation points, the aggregation shown in Figure 4d closely matches the true solution and significantly surpasses any single model used in the aggregation. We also observe that mere averaging across all models does not yield the same accuracy level. This highlights that significant gains in aggregation performance can be achieved by optimally incorporating covariance information both within and across models relative to the target.

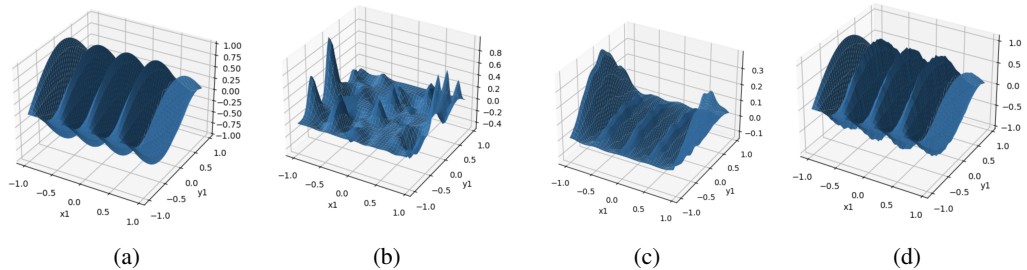

|  (a)  |  (b)  |  (c)  |  (d)  |

Figure 4: (a) Real solution of the PDE (b) One of the models aggregated (c) Uniform average of all models (d) Proposed aggregate equation 3

## C ANOTHER PATHOLOGICAL EXAMPLE: UNKNOWN REGION

In this second example illustrated in Figure 5 we select

$$\begin{cases} Y(x) &= 3\cos(2\pi x) \\ M_G(x) &= Y(x) + \epsilon(x) \quad \text{where } \epsilon(x) \sim \mathcal{N}(0, 0.3) \\ M_B(x) &= 3 \\ M_N(x) &= -3 \end{cases} \tag{90}$$

Write $M = (M_G, M_B, M_N)^T$ for the vector defined by the three models. It is common for aggregation methods, such as MoE, to constrain $\hat{\alpha}$ to be a convex combination, which we implement here by seeking an aggregate of the form $M_A(x) = Softmax(\hat{\nu}(x))^T M(x)$ where $Softmax(\hat{\nu}(x))_k := \frac{\exp(\hat{\nu}_k(x))}{\sum_{j=1}^{3} \exp(\hat{\nu}_j(x))}$. To identify the functions $\hat{\nu}_k$ we sample $N = 100$ points $X_i$ uniformly in $[-1, -0.5] \cup [0.5, 1]$ leaving the region $[-0.5, 0.5]$ with no data. We use a kernel method with an RBF kernel and loss 4, in which $\|\cdot\|_{\mathcal{H}}$ is the RKHS norm of $\kappa$. Using the representer theorem (Schölkopf et al., 2001), we reduce this loss to a finite-dimensional non-linear optimization problem and solve it using Gauss-Newton iterations (Wang, 2012). In this example, the good model is consistently better than the bad models over the entire dataset. Nevertheless, our aggregate fits the dataset perfectly by combining $M_B$ and $M_N$ instead of using $M_G$. In doing so, it fails to learn the quality of the models. Instead, it only interpolates the data, leading to poor performance outside the dataset despite having a model with good performance everywhere. This pathological behavior arises because the aggregate fails to account for model errors in its approximation of $Y$, even though it is a convex combination.

## D NON-INDEPENDENT MODEL ERRORS

### D.1 NEW FORMULAS FOR THE AGGREGATION

In section 3.3, we assume the models have independent errors. In general, this may not be true. Thus, we present here an extension to non-independent errors. Recall our approximation of the

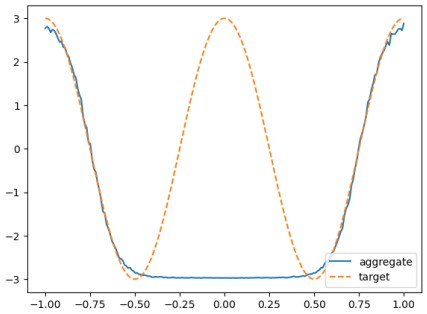

Figure 5: Pathological example (sec. 2.2)

covariance matrix $A(x)$:

$$\hat{A}(x) = P^T \begin{pmatrix} e^{\lambda_1(x)} & 0 & \cdots \\ 0 & \ddots & 0 \\ \vdots & 0 & e^{\lambda_n(x)} \end{pmatrix} P \quad \text{where} \begin{cases} PP^T = I_n \\ \lambda_i \text{ are regressors} \end{cases} \tag{91}$$

We may adapt the loss (12) to account for $P \neq I_n$, to get:

$$\lambda = \operatorname*{argmin}_{l \in \mathcal{H}} \sum_{i=1}^{N} \|P^T Diag(e^{l_k(X^i)})P - e^i(e^i)^T\|_F^2 + a\|l\|_{\mathcal{H}}^2 \tag{92}$$

$$= \operatorname*{argmin}_{l \in \mathcal{H}} \sum_{i=1}^{N} \sum_{k=1}^{n} \left( e^{l_k(X^i)} - (Pe^i)_k^2 \right)^2 + a\|l\|_{\mathcal{H}}^2 \tag{93}$$

With this formulation, we get a more general aggregation. In particular, despite summing to 1, the coefficients $\alpha(x)$ are not necessarily positive, giving added flexibility in case models have negative conditional correlations. The full Minimal Empirical Variance Aggregate (MEVA) with non-independent errors is given as:

$$\lambda = \operatorname*{argmin}_{l \in \mathcal{H}} \sum_{i=1}^{N} \sum_{k=1}^{n} \left( e^{l_k(X^i)} - (Pe^i)_k^2 \right)^2 + a\|l\|_{\mathcal{H}}^2 \tag{94}$$

$$\alpha(x)^T = \frac{\mathbb{1}^T P^T D(x)P}{\mathbb{1}^T P^T D(x)P\mathbb{1}} \quad \text{where } D(x) = Diag\left[\exp(-\lambda_k(x)), k = 1, ..., n\right] \tag{95}$$

$$M_A(x) = \sum_{i=1}^{n} \alpha_i(x)M_i(x) \tag{96}$$

### D.2 Choosing the matrix $P$

We have explored two options for the matrix $P$:

- $P = I$, the identity matrix, is the simplest choice.
- $P$ as the orthonormal basis that diagonalizes the empirical covariance matrix $C := \frac{1}{N}\sum_{i=1}^{N} e_i e_i^T$.

In the experiments presented below, we only show results for $P = I$. This choice is straightforward and delivers accuracy comparable to or better than using the eigenvectors of $C$.

We also observed that using the empirical covariance matrix can render the method unstable if the models are highly correlated. Indeed, consider, for instance, two models whose errors are significantly correlated, leading to the following empirical covariance matrix:

$$C := \begin{pmatrix} 1.1 & 0.9 \\ 0.9 & 1 \end{pmatrix} \tag{97}$$

In this case, the second eigenvector of $C$, $P_2 \approx \frac{1}{\sqrt{2}}(0.97, -1.02)$, results in $\mathbb{1}^T P_2 \approx -0.04$, which effectively subtracts the two models. This eigenvector is associated with a small eigenvalue ($0.15$), indicating that the difference between the two models has a low variance on average. Since equation (7) depends on the inverse of the estimated covariance matrix, a small estimated variance eigenvalue $e^{\lambda_2(x)}$ leads to $\alpha(x)$ having large coefficients with opposing signs. In this scenario, we observed that the aggregation method can become sensitive to small errors and thereby suffer from poor generalization.

## E  AN ALTERNATIVE, DIRECT MINIMAL EMPIRICAL VARIANCE AGGREGATION LOSS

Here, we propose an alternate end-to-end loss for the minimal empirical variance aggregation (MEVA). In section 3.1, we observed that our approach can be interpreted as minimizing the aggregate's variance, using an unbiased combination $\alpha$ s.t. $\mathbb{1}^T \alpha = 1$. Under the model (6), $Cov[M(x)] = A(x)$, so our MVA is such that

$$\alpha(x) = \underset{\nu \in \mathbb{R}^n \sum_{i=1}^n \nu_i = 1}{\operatorname{argmin}} \nu^T A(x) \nu \tag{98}$$

Instead of estimating $A(x)$ and inverting this estimation to obtain the minimal variance aggregate as in (7), we propose using an empirical version of the above loss. Specifically, let the empirical covariance matrices $A_i$ be:

$$A_i := \begin{cases} P^T Diag((Pe^i)_k^2, k = 1..,n)P & \text{for general } P \\ Diag((Y^i - M_k(X^i))^2, k = 1..,n) & \text{if } P = I_n \end{cases} \tag{99}$$

Write $\mathcal{H}_1$ for a normed subspace of the space of unbiased aggregators $\{u : x \mapsto u(x) \in \mathbb{R}^n \text{ s.t. } \sum_{i=1}^n u_i(x) = 1, \forall x\}$. Then, the proposed alternative approach identifies an aggregator $\tilde{\alpha}$ by minimizing the sample variance of the aggregate:

$$\tilde{\alpha} = \underset{u \in \mathcal{H}_1}{\operatorname{argmin}} \sum_{i=1}^N u(X^i)^T A_i u(X^i) + a\|u\|_{\mathcal{H}_1}^2 \tag{100}$$

We employ this strategy in the Boston dataset example (section 4.1). Here, $\mathcal{H}_1$ is a set of vector-valued functions parameterized by a neural network, where the output is constrained to have positive entries summing to one via a softmax layer. The term $\|u\|_{\mathcal{H}_1}^2$ is an $L^2$-norm regularizer applied to the network's weights and biases. This method is both straightforward and practical.

## F  ADDING A BIAS CORRECTION

Assuming the models $M(x)$ to be unbiased is necessary to obtain a linear aggregate, as adding a bias correction will result in an affine aggregation of the form $M_A(x) = \alpha^T(x)M(x) + \beta(x)$. This assumption may, however, be violated in practice, and in this case, bias will be interpreted as variance. In a context with limited data and little knowledge of the models and target, it would be difficult to know if an error can be attributed to bias or variance. In the case where it is clear models are biased and interpreting it as variance hurts accuracy, we may modify our method by accounting for the bias

$$M(x) = Y(x)\mathbb{1} + \tilde{Z} \text{ where } \begin{cases} \mathbb{E}[\tilde{Z}(x)] = \mu(x) \\ Cov[\tilde{Z}(x)] = A(x) \end{cases} \tag{101}$$

In such a model, we must, in addition to $\lambda$, train $\mu$. Using the definition of $\tilde{Z}$, we know the loss for $\mu$ is of the form $\sum_{i=1}^N (\mu(X^i) - e^i)^T A^{-1}(X^i)(\mu(X^i) - e^i)$. Taking $P = I$, this bias-corrected

aggregation would be:

$$\tilde{\lambda}, \tilde{\mu} = \underset{l,m \in \mathcal{H}_1 \times \mathcal{H}_2}{\mathrm{argmin}} \sum_{i=1}^{N} \sum_{k=1}^{n} \left[ \left( e^{l_k(X^i)} - (e_k^i - m_k(X^i))^2 \right)^2 + e^{-l_k(X^i)} \left( m_k(X^i) - e_k^i \right)^2 \right] \tag{102}$$

$$+ a\|l\|_{\mathcal{H}_1}^2 + b\|m\|_{\mathcal{H}_2}^2 \tag{103}$$

$$\tilde{\alpha}(x) = Softmax(-\tilde{\lambda}(x)) \tag{104}$$

$$\tilde{M}_A(x) = \sum_{i=1}^{n} \tilde{\alpha}_i(x)(M_i(x) - \tilde{\mu}_i(x)) \tag{105}$$

where $\mathcal{H}_1, \mathcal{H}_2$ are two spaces of functions used to approximate the log-eigenvalues of $A(x)$ and the bias of the models.

In the Boston housing dataset example in section 4.1, we tried to implement this bias-corrected aggregation, and it did not yield better results than the simpler method MEVA. While we do not prescribe its use in general, we did not consider this model in the rest of the paper.

# G AGGREGATION USING VECTOR-VALUED GAUSSIAN PROCESSES AND MATRIX-VALUED KERNELS

We will now model aggregation as vector-valued Gaussian process regression to elucidate the pathologies observed in the two abovementioned examples. We start with a brief reminder on this vector/matrix-valued generalization of GPs/kernels (Alvarez et al., 2012), before computing the MEA in the GP context.

## G.1 VECTOR-VALUED GAUSSIAN PROCESS AND MATRIX-VALUED KERNEL

This section is a reminder on vector/matrix-valued GPs/kernels Alvarez et al. (2012). We define a matrix-valued kernel as a function $K : \mathcal{X} \times \mathcal{X} \to \mathbb{R}^{n \times n}$ such that,

- $\forall x, x' \in \mathcal{X}, K(x, x') = K(x', x)^T$
- $\forall x_1, .., x_N \in \mathcal{X}, \forall y_1, .., y_N \in \mathbb{R}^n,$

$$\sum_{1 \le i,j \le N} y_i^T K(x_i, x_j) y_j \ge 0 \tag{106}$$

This generalizes the positivity and symmetry conditions for defining a kernel. Each matrix-valued kernel uniquely defines a zero-mean vector-valued Gaussian process $\xi$ such that :

- $\forall x \in \mathcal{X}, \xi(x) \sim \mathcal{N}(0, K(x, x))$
- $\forall x, x' \in \mathcal{X}, Cov\left[\xi(x), \xi(x')\right] = K(x, x')$

One interesting example of such a vector-valued Gaussian process is the case of $n$ independent Gaussian processes, corresponding to $K$ being diagonal. One can then understand general vector-valued GP as adding correlation to the coordinates, each being its own GP.

## G.2 MEA WITH MATRIX-VALUED KERNELS

We choose a matrix-valued kernel $K : \mathcal{X} \times \mathcal{X} \to \mathbb{R}^{n \times n}$, and write $\mathcal{H}_K$ for its associated Reproducing Kernel Hilbert Space (RKHS). We get the MEEA by solving equation 4 (see Appendix H):

$$M_A(x) = \hat{\alpha}_E(x)^T M(x) = \tilde{k}(x, X)(\tilde{k}(X, X) + \lambda I)^{-1} Y \tag{107}$$

where $\tilde{k}(x, y) = M(x)^T K(x, y) M(y)$. This is also the GP regressor $M_A = \mathbb{E}\left[\xi \mid \xi(X) = Y + \mathcal{N}(0, \lambda I)\right]$ with prior $\xi \sim \mathcal{N}(0, \tilde{k})$. This observation is crucial as it shows that the model values are used as a feature for the regression, as expected. However, it demonstrates that

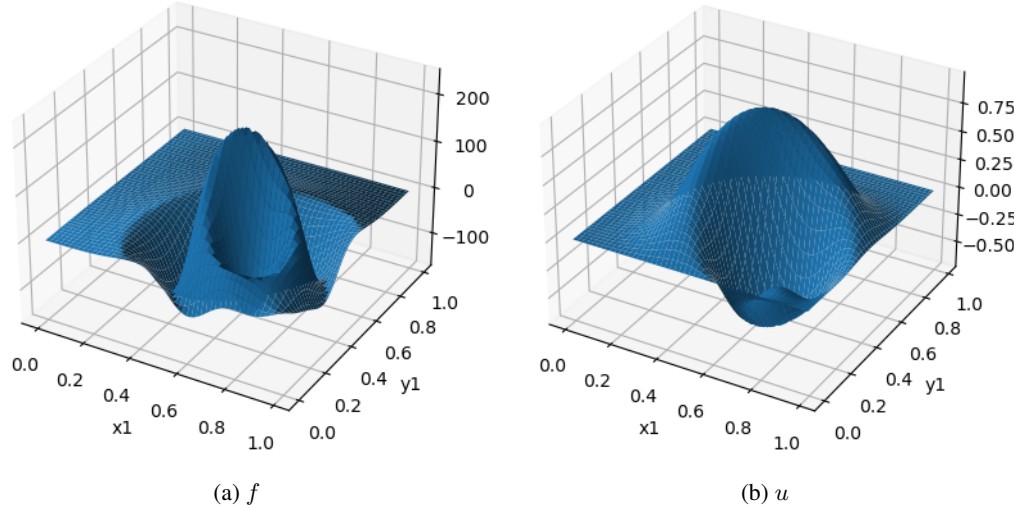

(a) $f$           (b) $u$

Figure 6: A pair $(f, u)$ s.t. $-\Delta u = f$ and $u(\partial\Omega) = 0$, sampled using (18)

the Mean Empirical Error Aggregate (MEEA) only tries to regress $Y$ without considering the underlying models' accuracy. This observation is counterintuitive as we expect model aggregation to leverage the individual models to approximate the target function better rather than merely regressing directly to the target. This behavior is particularly evident in pathological examples, illustrating the limitations of the direct regression approach.

## H    MINIMIZING THE LOSS IN SECTION G

We aim to minimize the following optimization problem:

$$\hat{\alpha} = \operatorname*{argmin}_{\alpha \in \mathcal{H}_K} \frac{1}{N} \sum_{i=1}^{N} \left[ Y^i(x_i) - \alpha(x_i)^T M^i(x_i) \right]^2 + \gamma \|\alpha\|_{\mathcal{H}_K}^2 \tag{108}$$

Applying the representer theorem, we deduce that the solution $\hat{\alpha}$ can be represented as $\hat{\alpha} = (K(\cdot, x_1), \ldots, K(\cdot, x_N))V$, where $V \in \mathbb{R}^{nN}$. Additionally, we define the matrix $\mathcal{M} \in \mathbb{R}^{N \times nN}$ as follows:

$$\mathcal{M} = \begin{pmatrix} M_1 \\ \vdots \\ M_N \end{pmatrix}, \tag{109}$$

where each $\mathcal{M}_i = (M(x_i)^T K(x_i, x_1), \ldots, M(x_i)^T K(x_i, x_N))$. The optimal vector $V$ is then given by:

$$V = \operatorname*{argmin}_{v \in \mathbb{R}^{nN}} \|Y - \mathcal{M}v\|^2 + \lambda v^T \mathcal{K}v \tag{110}$$

where $\mathcal{K}$ is a block matrix such that $\mathcal{K}_{ij} = K(x_i, x_j)$ for $1 \leq i, j \leq N$. Define the matrix $D \in \mathbb{R}^{N \times nN}$ as:

$$D = \begin{pmatrix} M(x_1)^T & 0 & \ldots & 0 \\ 0 & M(x_2)^T & \ldots & \ldots \\ \vdots & \ldots & \ddots & \vdots \\ 0 & \ldots & \ldots & M(x_N)^T \end{pmatrix} \tag{111}$$

Observe that $\mathcal{M} = D\mathcal{K}$. Using the matrix identity:

$$(P^{-1} + B^T R^{-1} B)^{-1} B^T R^{-1} = PB^T (BPB^T + R)^{-1} \tag{112}$$

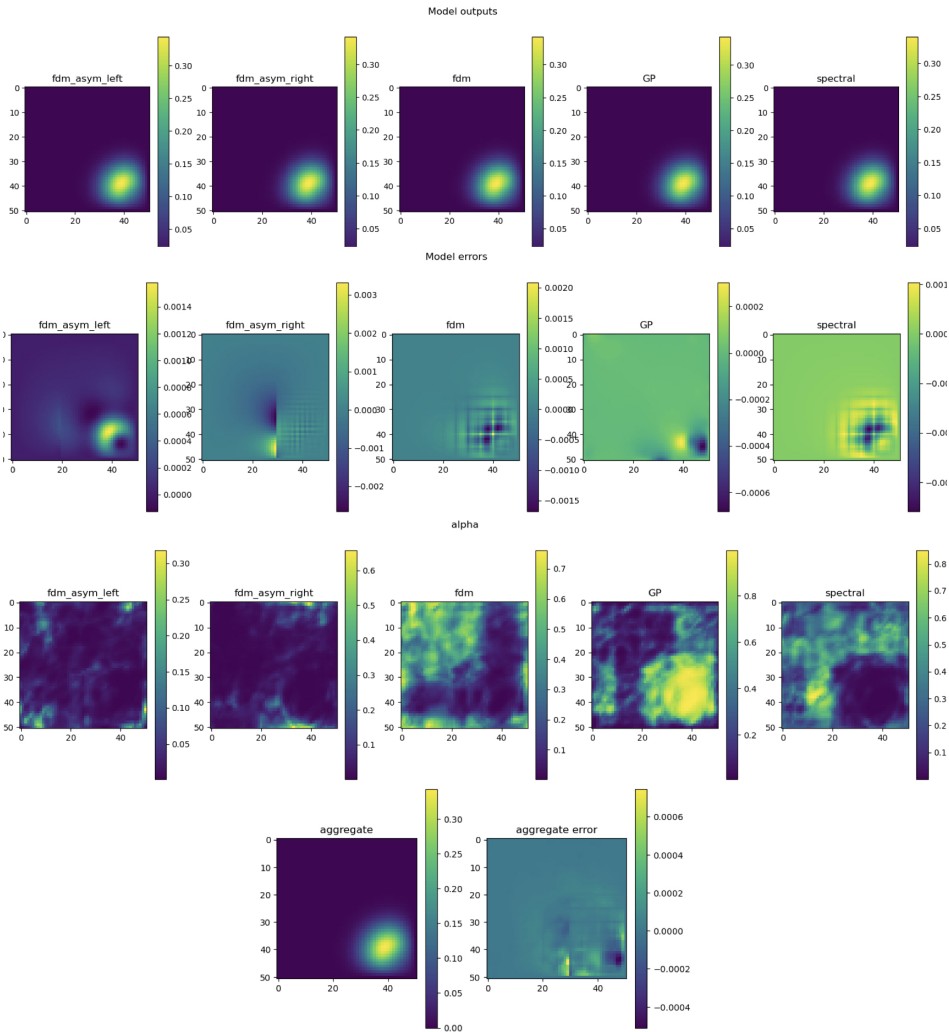

Figure 7: Example of aggregation for the Laplace equation (section 4.3). Line 1: Outputs of the different models for a given input $f$. Line 2: Prediction errors for each model. Line 3: Values of $\alpha$. Line 4: Aggregate and its error

where $P^{-1} = \lambda K$, $B = \mathcal{M}$, and $R = I$, we derive:

$$V = \frac{1}{\lambda} K^{-1} \mathcal{M}^T \left( \frac{1}{\lambda} \mathcal{M} K^{-1} \mathcal{M}^T + I \right)^{-1} Y \tag{113}$$

$$= D^T (D\mathcal{K}D^T + I)^{-1} Y \tag{114}$$

It follows that $(D\mathcal{K}D^T)_{ij} = M^T(x_i) K(x_i, x_j) M(x_j) = \tilde{k}(x_i, x_j)$. Thus, the final model prediction at a new point $x$ is given by:

$$M_A(x) = M(x)^T K(x, X) V \tag{115}$$

where $K(x, X) = (K(x, x_1), \dots, K(x, x_N))$ so that $\hat{\alpha}(x) = K(x, X)V$. By identifying that $M(x)^T K(x, X) D^T = \tilde{k}(x, X)$, we can conclude:

$$M_A(x) = \tilde{k}(x, X)(\tilde{k}(X, X) + \lambda I)^{-1} Y \tag{116}$$

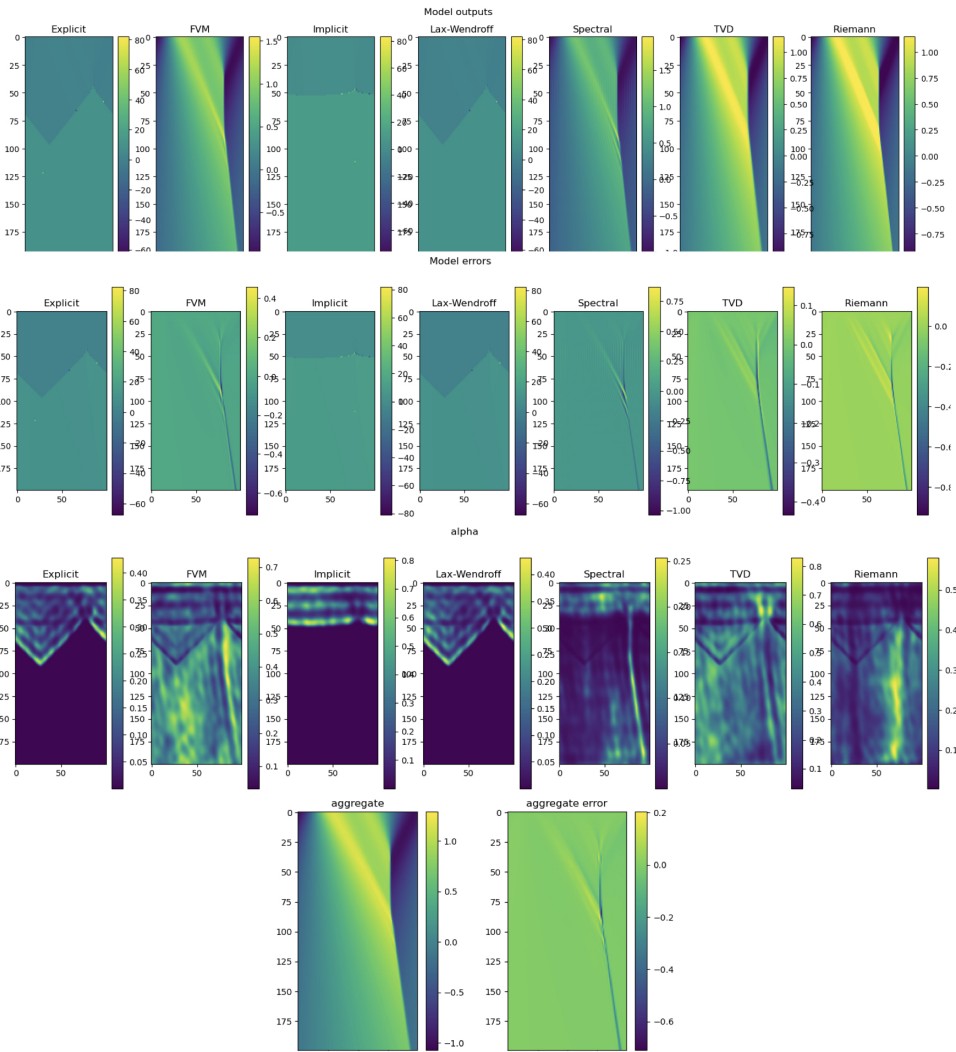

Figure 8: Example of aggregation for Burger's equation (section 4.4). Line 1: Outputs of the different models for a given input $f$. Line 2: Prediction errors for each model. Line 3: Corresponding values of $\alpha$. Line 4: Aggregate and its error. Note that the explicit, implicit, and Lax-Wendroff methods all diverged, while the spectral method exhibited spurious oscillations. In contrast, the aggregated result avoids these issues.

