# OpenReview forum: "Minimal Variance Model Aggregation: A principled, non-intrusive, and versatile integration of black box models"
_ICLR.cc/2025/Conference — ICLR 2025 Poster_

### Official Review · Reviewer_aX4C · 2024-10-17

**Soundness:** 3
**Presentation:** 3
**Contribution:** 3
**Rating:** 6
**Confidence:** 3

**Summary:**

This paper focuses on model aggregation by minimizing empirical variance rather than empirical bias, which can lead to better generalization. By making appropriate assumptions about the error model, the authors present a tractable aggregation scheme using nonparametric methods. They evaluate their approach on a housing dataset and two PDE solver problems, demonstrating its accuracy and effectiveness.

**Strengths:**

1. This paper highlights the drawbacks of current aggregation approaches that focus on minimizing prediction error, using extensive discussions and interpretable examples to provide valuable insights into model aggregation.
2. The proposed Minimal Empirical Variance Aggregation method is intuitive and performs effectively in certain model aggregation tasks.
3. The implementation on PDE solvers is both interesting and practically useful.

**Weaknesses:**

1. There is a lack of rigorous theoretical discussion on the superiority of MEVA over MEA. Including some theoretical investigations would be preferable. Besides, the implementation of MEVA requires certain approximations. Some theoretical guarantee for the final performance of MEVA towards MVA should be considered.
2. The derivations of MVA and MEVA rely heavily on assumptions about the error model, which may not hold in practice. The authors should address the issue of misspecification in the paper.
3. The experiments using real-world data are not comprehensive, as only a single housing dataset is evaluated. It is necessary for the authors to include more experiments to further demonstrate MEVA's performance if the paper is accepted.
4. Although I am not familiar with the background of model aggregation, I noticed the absence of a literature review and comparison with relevant benchmarks. The cited literature in the related work section consists mainly of reviews and some unrelated papers in scientific computing, and there is no specific model aggregation approaches compared in the real-data experiments. More comparing benchmarks should be discussed and evaluated in the paper. For example, in econometrics, there is extensive literature on model averaging [1,2], which also uses linear combinations of models to improve predictions. The authors should discuss these approaches and include comparative experiments.

[1] Hansen, B. E. (2007). Least squares model averaging. Econometrica, 75(4), 1175-1189.

[2] Zhang, X., & Liu, C. A. (2023). Model averaging prediction by K-fold cross-validation. Journal of Econometrics, 235(1), 280-301.

**Questions:**

1. Figure 2. It appears that MEVA does not outperform the random forest model in the housing data experiment. Can the authors explain the reasons for this outcome? And I have a minor comment on Figure 2. The labels in x-axis are too small to see them clearly. It is better to arrange this.
2. Can the authors discuss the robustness of the approximation of $A(x)$ mentioned in Line 258? Is it reasonable to expect this approximation to hold in practice, and what are the potential impacts if the underlying assumptions are incorrect? A discussion on this issue, along with references to relevant literature, would be appreciated.

---

> ### Author Response · Authors · 2024-11-26
> **Comments on the weaknesses you pointed out**
>
> Thank you for your assessment. To comment on the weaknesses you pointed out:
>
> **Theoretical guarantees**: In the new version of the manuscript, we prove a theorem showing that MEVA's loss converges much faster to its limiting value than MEEA. Thus, despite making an error (which we argue is small in model aggregation) compared to MEA by adding extra constraint, this regularizes the aggregation and provides more robust estimates. In our context where $N$ datapoints are used to extrapolate to all $x$, which we identify as low data regime in general, this faster convergence ensures MEVA's superiority in practice.
>
> **Impact of bias**: Our theoretical results take into account this misspecification and show MEVA's fast convergence even when there is bias. Many existing methods that employ pointwise linear aggregation rely on the heuristic that weights sum to 1. In our work, this idea naturally stems from the unbiasedness assumption, providing both a theoretical justification and a framework for improving the method when this assumption is violated. Appendix D specifically details a bias correction procedure for such cases.
> In practice, accounting for bias has not significantly influenced performance in our experiments. We attribute this to the minimal assumptions placed on the error covariance, allowing the aggregation process to interpret all errors as variance. Additionally, the distinction between bias and variance may often be more of a theoretical construct than a practical issue. For instance, in the PDE example, all methods are deterministic, meaning that all errors are inherently biased. Despite this, our method performs robustly, even in this misspecified setting.
> Similar to how linear regression assumes Gaussian errors but remains effective across a wide range of scenarios, our bias assumption serves to justify the aggregation formula and provide a theoretical foundation. However, the method demonstrates robustness in practice and does not necessarily fail when this assumption is not strictly met.
>
> **Complexity of examples**: We acknowledge the limited inclusion of real-world examples in this work. However, this paper represents the first attempt to extend aggregation techniques beyond simple regression settings, making it inherently exploratory. Our focus was on developing and validating the method in controlled environments.
> We are actively pursuing applications in scientific computing, particularly in fields such as weather and climate prediction, to tackle real-world problems and demonstrate the broader applicability of our approach.
>
> **Benchmarking**: Upon further review of existing methods, we recognize that the current state-of-the-art is well established in regression tasks involving surrogate models, where the goal is to recover known analytic functions. We have updated the related works section to reflect this observation. Notably, these methods can be interpreted as specific cases of MEVA, using a predefined regressor for the error.
> While the Boston Housing dataset aligns broadly with the regression setting, its inclusion in our work primarily serves as a sanity check to validate whether aggregation improves the performance of the models being combined. The core of our experimental contribution lies in the novel application of model aggregation to PDE solvers and operator learning. In these contexts, benchmarking is inherently challenging due to the lack of existing comparable work. These examples better capture the complexities of real-world scientific computing problems, which we aim to address in future applications.

---

> ### Author Response · Authors · 2024-11-26
> **To answer your questions**
>
> To answer your questions:
>
> **Performance on the Boston Housing dataset**: We have updated the Boston Housing dataset example with better hyperparameters and averaging over multiple seeds. Note that ensemble learning techniques such as random forests and gradient boosting are state-of-the-art for regression tasks and are optimized for this setting, it is unsurprising that they remain difficult to surpass with the same data.
> That said, the data science setting is not the primary focus of our method or where it is most warranted. The inclusion of this experiment was intended to validate our approach by demonstrating that aggregation improves model performance. However, in practice, there is little reason to use non-trainable models in this context. In data science settings, our method serves as an effective aggregation tool, but trainable ensemble techniques like random forests and boosting remain better suited for this domain.
>
> **Robusteness of approximation of $A$**: Most methods in the literature employing pointwise linear aggregation rely on convex combinations as a heuristic, effectively approximating  A  as diagonal. In our work, we simplify the problem by defining $A=P Diag(\lambda_i)P^T$ , which keeps the regression tractable.
> This approach can be compared to specifying a matrix-valued kernel in multi-dimensional Gaussian Process (GP) regression, as outlined in Appendix E. In general, defining a covariance structure across the entire domain  x  is a complex task, akin to recovering the full correlation structure in our setting. Commonly, GPs are assumed to be either independent or separable, where the covariance is represented as the product of a 1D kernel and a positive semi-definite (PSD) matrix  Q  [1]. To introduce additional complexity, kernels may be combined as described in Theorem 3 [2]:
>
> $K(x,y) =\sum_{i=1}^p k_i(x,y) Q_i$
>
> Our method is analogous, specifying the covariance structure as:
>
> $A(x)=\sum_{i=1}^n e^{\lambda_i(x)}P_iP_i^T\text{ where }P_i\text{ are the columns of }P$
>
> Since  A  must remain PSD, one potential approach when the approximation does not hold is to regress the Cholesky decomposition of  A . However, this would come at the cost of having to regress a quadratic number of coefficients, significantly increasing complexity.
> [1] Sindhwani, V., Quang, M. H., & Lozano, A. C. (2012, October). Scalable Matrix-valued Kernel Learning for High-dimensional Nonlinear Multivariate Regression and Granger Causality. Retrieved from http://arxiv.org/abs/1210.4792
> [2] Kadri, H., Duflos, E., Preux, P., Canu, S., Rakotomamonjy, A., & Audiffren, J. (2016, November). Operator-valued Kernels for Learning from Functional Response Data. Retrieved from http://arxiv.org/abs/1510.08231

---

> ### Author Response · Authors · 2024-11-26
> **New version of the manuscript**
>
> We have made several changes to the manuscript to improve the article.
>
> We have modified the title, abstract, and introduction to better highlight the contributions of our work. In particular, we have revised the related work section and added a clear contributions section.
> We have proved a theorem to address concerns about the well-foundedness of the method. This theorem justifies the superiority of MEVA over MEEA by showing it has a faster convergence rate, making it more robustly estimated. We identify our setting as a low data regime, making robust estimation key for accuracy.
> We have reworked the Boston housing dataset example. Its role as a sanity check for the method has been emphasized. We also noticed the small data set size made results highly variable. We now average results over many seeds, and after better hyperparameter choice, our method shows a 2% overall improvement.

---

> > ### Comment · Reviewer_aX4C · 2024-11-27
> >
> > Thank you for your effort to improve the paper. Most of my concerns have been well addressed. And I have updated my score to 6. Good luck! And one minor comment: Page 20, Line 1041, the figure number is not correctly shown.

---

> ### Author Response · Authors · 2024-11-27
>
> Thank you for your constructive feedback, which helped us improve the paper, and thank you for the support!
>
> (figure reference was updated)

---

### Official Review · Reviewer_ppzU · 2024-11-03

**Soundness:** 3
**Presentation:** 4
**Contribution:** 3
**Rating:** 6
**Confidence:** 3

**Summary:**

This paper presents a data-driven framework designed to consolidate predictions from various models, thereby improving overall accuracy by capitalizing on their unique strengths. This approach is non-intrusive and model-agnostic, treating the contributing models as black boxes and allowing for the integration of outputs from a wide range of methodologies, such as machine learning techniques and conventional numerical solvers. The authors introduce a point-wise linear aggregation method and investigate two strategies for optimizing this aggregation:

The first, Minimal Error Aggregation (MEA), aims to directly minimize prediction errors, while the second, Minimal Variance Aggregation (MVA), seeks to lower variance by estimating the errors of the models. Although MEA is theoretically optimal when the correlations between models and the target are perfectly understood, its empirical counterpart, Minimal Empirical Error Aggregation (MEEA), frequently encounters issues with overfitting due to inaccuracies in correlation estimates. The authors illustrate that the empirical variant of MVA, termed Minimal Empirical Variance Aggregation (MEVA), consistently surpasses MEEA by effectively estimating model errors and aggregating the results. This method mitigates the risk of overfitting and improves generalization capabilities. The efficacy of MEVA is validated through various applications, including the Boston Housing dataset and the resolution of partial differential equations (PDEs) such as the Laplace and Burgers' equations. In these tests, MEVA demonstrates superior performance compared to individual models and MEEA, leading to a more resilient and precise aggregate model.

**Strengths:**

The paper introduces an innovative methodology for model aggregation that shifts the emphasis from the conventional goal of minimizing empirical error to the more nuanced objective of minimizing empirical variance. This approach is relatively uncommon in existing literature. Traditional ensemble techniques, such as bagging and boosting, primarily concentrate on reducing variance or bias through averaging or additive strategies. In contrast, this study highlights the importance of variance minimization via error estimation, a perspective that has not been extensively investigated.

The theoretical underpinnings of the proposed methods are robust, featuring comprehensive derivations and justifications. The authors conduct an in-depth examination of scenarios where the Model Error Estimation Aggregation (MEEA) falters while the Model Error Variance Aggregation (MEVA) excels, offering valuable insights into the benefits of focusing on variance minimization. This discussion is pertinent to the well-documented challenges of overfitting associated with empirical risk minimization, where models that prioritize empirical error reduction may struggle with generalization.

The organization and clarity of the paper are commendable, as it effectively presents complex ideas in an accessible manner. Necessary definitions are provided, and the step-by-step explanations aid in comprehension. The use of illustrative examples, figures, and comprehensive descriptions of experimental methodologies further enhances the work's readability and transparency.

By illustrating that a focus on minimizing empirical variance can yield superior aggregation performance compared to merely minimizing empirical error, this paper makes a good contribution to the field of ensemble learning. The capacity to aggregate a variety of models, including those derived from different methodologies and domains, underscores the versatility of this approach. This has important implications for numerous fields where the integration of multiple predictive models is advantageous, such as climate modeling, where ensemble methods are employed but often rely on simplistic aggregation techniques.

**Weaknesses:**

The methodology is predicated on the premise that individual models serve as unbiased estimators of the target variable. However, in real-world applications, models may demonstrate bias stemming from systematic errors or incorrect model specifications. This concern is well-documented in the context of model averaging, as highlighted by Claeskens and Hjort (2008), where the presence of biased models can detrimentally influence the overall aggregation. Although the paper acknowledges this limitation, it does not delve deeply into the implications of such assumption violations on the aggregation process or propose modifications to accommodate biased models.

2. The experiments conducted, while illustrating the efficacy of MEVA in certain scenarios, exhibit a degree of limitation in their breadth. The Boston Housing dataset, being relatively small, may not adequately capture the complexities associated with larger and more intricate datasets. Furthermore, while the PDE examples are pertinent, they could be enhanced by incorporating additional real-world applications to better demonstrate the method's applicability across various contexts. Recent advancements in operator learning, as discussed by Kovachki et al. (2023), tackle complex PDEs in higher dimensions, and evaluating MEVA within these frameworks would bolster the validity of its claims.

3. In the experiment utilizing the Boston Housing dataset, the enhancements offered by MEVA over alternative methods appear to be modest, particularly regarding mean squared error (MSE). This observation prompts a reevaluation of the practical significance of the method's purported benefits in conventional machine learning tasks. Established ensemble techniques, such as gradient boosting, have demonstrated substantial improvements in these areas, suggesting that a more comprehensive comparison of MEVA's performance against these well-regarded methods would be advantageous.

**Questions:**

The inquiry into the handling of biased models raises significant questions regarding the assumption of individual model impartiality. It is essential to explore whether the MEVA method can be adapted to include biased models and what specific alterations would be required for such an extension. Additionally, it is important to consider how the existence of bias within the models might influence the overall performance of the aggregation process.

The presence of correlated model errors in practical applications necessitates an understanding of how MEVA addresses these correlations. It is pertinent to evaluate the sensitivity of aggregation performance to such error correlations and to consider whether incorporating off-diagonal elements in the covariance matrix could enhance the model's robustness.

A comparative analysis of MEVA against other established aggregation methods, such as stacking, weighted averaging, or contemporary ensemble techniques, would provide valuable insights into its relative performance and practical benefits.

Lastly, the selection of function spaces, such as Reproducing Kernel Hilbert Spaces (RKHS), and the tuning of regularization parameters are critical components of the method. Practical guidelines or heuristics for these selections, as well as an assessment of how sensitive the method's performance is to these hyperparameters, would be beneficial for practitioners.

---

> ### Author Response · Authors · 2024-11-26
>
> Thank you for your assessment. To comment on the weaknesses you pointed out:
>
> **Impact of bias**:  Many existing methods that employ pointwise linear aggregation rely on the heuristic that the weights sum to 1. In our work, this natural idea is derived directly from the unbiasedness assumption, providing a theoretical justification and a framework for improvement if this assumption is violated. Specifically, Appendix D outlines a bias correction procedure for such scenarios.
> Notably, accounting for bias did not significantly affect performance in our experiments. We attribute this to:
> - The fast convergence of the loss of MEVA to its limiting value, at rate $\frac{1}{N}$ (compared to the Monte Carlo rate $\frac{1}{\sqrt{N}}$). This discussion is part of the larger theoretical contributions we added to our manuscript, which prove the superiority of MEVA over MEEA.
> - The minimal assumptions placed on the error covariance, which enables the aggregation process to interpret and adapt to all errors as variance. This observation reinforces the robustness of the method in practice. These considerations have been clarified in the section describing MEVA.
>
> **Complexity of examples**: We acknowledge that the operator learning literature now includes very complex examples, particularly in higher-dimensional settings. However, this work represents the first attempt to extend model aggregation beyond simple regression tasks, making it inherently exploratory. Our focus was on establishing the method and demonstrating its potential in controlled settings.
> We are actively seeking applications in scientific computing, particularly in fields like weather and climate prediction, to address real-world challenges and further validate our approach.
>
> **The Data Science Setting**: The data science setting is not the primary focus of our method, nor is it where our approach is most warranted. While the question of aggregating existing models is valid and was addressed in the paper as a means to validate our approach, there is little practical reason to use non-trainable models in this context. In data science, our method effectively aggregates models and improves upon their individual performance. However, treating the models as trainable would likely yield even better results, as ensemble learning techniques like boosting have already been thoroughly explored and perform exceptionally well in this domain. It is not our goal to compete with such established methods in this setting.
> On the other hand, structured problems with specific design constraints, such as those encountered in scientific computing, particularly PDE solving, are better suited for our method. In these settings, non-trainable models or models with distinct drawbacks are common, making aggregation critical. The data science experiment included in the paper serves as a sanity check using a known benchmark, but our primary contribution and the applicability of our methods lie in addressing problems of the PDE kind.

---

> ### Author Response · Authors · 2024-11-26
> **To answer your questions**
>
> **Bias**: Our theoretical results take into account this misspecification and show MEVA's fast convergence even when there is bias.
> The extension to biased models is addressed in Appendix D, and we have added a clearer pointer to this appendix in the presentation of MEVA within the paper. It is worth noting that the distinction between bias and variance may often be more of a theoretical construct than a practical concern. For instance, in the PDE example, all methods are deterministic, meaning that all errors are, by definition, attributed to bias. Despite this, our method still performs well.
> Similar to how linear regression assumes Gaussian errors but remains effective in a wide range of scenarios, our bias assumption serves to justify the aggregation formula and provide a theoretical foundation. However, the method is robust in both theory and practice and does not necessarily fail when this assumption is not strictly met.
>
> **Error correlations**: Correlated model errors are treated as a generalization in Appendix B. In the examples we considered, this complexification did not yield better results.
>
> **Comparison with other techniques**:  In all our experiments, we compare our approach with both MEEA (which can be seen as a variant of the popular Nested-Kriging approach) and the simple average of models. Since we do not prescribe a specific regression method for the aggregation process, weighted averages (which are common in the literature) are naturally included as a possible instantiation of our framework. The choice of an appropriate regressor depends on the specific application, allowing flexibility in adapting the method to different scenarios.
> Furthermore, our work represents the first application of model aggregation in the context of operator learning. As such, there are no existing methods to directly compare against for the PDE example, which highlights the novelty of our approach in this domain.
>
> **Practical guidelines**: While we describe the class of regressor used (FNO or Matern kernel), practical considerations are left in the GitHub of the paper, where the full detail of the implementation is detailed. We note that we used the classical ML approach to tackle this supervised learning approach, and in particular, the PDE examples required very little tuning.

---

> ### Author Response · Authors · 2024-11-26
> **New version of the manuscript**
>
> We have made several changes to the manuscript to improve the article.
>
> We have modified the title, abstract, and introduction to better highlight the contributions of our work. In particular, we have revised the related work section and added a clear contributions section.
> We have proved a theorem to address concerns about the well-foundedness of the method. This theorem justifies the superiority of MEVA over MEEA by showing it has a faster convergence rate, making it more robustly estimated. We identify our setting as a low data regime, making robust estimation key for accuracy.
> We have reworked the Boston housing dataset example. Its role as a sanity check for the method has been emphasized. We also noticed the small data set size made results highly variable. We now average results over many seeds, and after better hyperparameter choice, our method shows a 2% overall improvement.

---

> ### Comment · Reviewer_ppzU · 2024-12-03
>
> Thank you for your detailed responses and the thoughtful revisions to the manuscript. The clarifications on bias robustness and the focus on MEVA’s strengths in PDE applications are appreciated and add depth to the paper. I believe that the scores given earlier accurately reflect the strengths and contributions of the work as it stands.

---

### Official Review · Reviewer_k7LR · 2024-11-05

**Soundness:** 3
**Presentation:** 3
**Contribution:** 3
**Rating:** 6
**Confidence:** 2

**Summary:**

This paper compares two different approaches to model aggregation: the first approach is called Minimal Empirical Error Aggregation (MEEA), which minimizes (over $a$): $\sum_{i=1}^N\left|Y^i-a\left(X^i\right)^T M\left(X^i\right)\right|^2+\lambda\|a\|_{\mathcal{H}}^2$, where $(X^i, Y^i)$ are $N$ data points, $\mathcal{H}$ is a set of functions used for the approximation, and $\lambda \geq 0$. In this approach, models $M\left(X^i\right)$ are treated as if they are features in regression. Section 3 gives pathological examples that show that this approach may produce an unreasonable aggregation. The second approach called Minimal Empirical Variance Aggregation (MEVA) has the same form of linear weights but $\alpha(x)$ is obtained differently: i.e., $\alpha(x)^T M(x)$, where the $i$-th element of $\alpha(x)$ equals $\exp(-\lambda_i(x))/[\sum \exp(-\lambda_k(x))]$, where summation is over $k$. To obtain $\lambda_i$, first write $e^i$ for the vector with entries defined as the sample errors $\left(e^i\right)_k=M_k\left(X^i\right)-Y\left(X^i\right)$. Then, $\lambda$ minimizes (over $l$): $\sum \sum \left(\exp(l_k\left(X^i\right))-\left(e^i\right)_k^2\right)^2$ plus some penalty term, where double summation is over $i$ and $k$. The numerical experiments consider aggregation using the Boston housing dataset as well as aggregation of PDE solvers.

**Strengths:**

- MEVA seems an interesting idea and can be potentially impactful when model aggregation is necessary.
- It sounds neat to cast numerical PDE as combining Partial Differential Equation (PDE) solvers through model aggregation.
- The paper is professionally edited and written.

**Weaknesses:**

- It is unclear what the benchmark is in the literature. The method called MEEA may not be commonly used in the literature and there is no clear sense that this paper improves the current state-of-the-art.
- There is no theoretical result that guarantees the superior performance of MEVA relative to MEEA or simple averaging (that is, $\alpha(x) = 1/n$ uniformly over $x$).
- There is a hyper parameter to be tuned (e.g., the penalization parameter $a$ in equation (13)), but it is unclear how to choose it.

**Questions:**

- Perhaps I missed this, but it is unclear how to specify the $l(\cdot)$ function in equation (13). It would be useful to provide a concrete example of this, after stating equation (13).
- What are the most popular or best performing methods in PDE solvers? It might be good to indicate how well the propose method works relative to the most popular or best performing methods in practice.
- Would it be possible to provide computational speed or difficulties as aggregation would take more time than using only one individual model?

---

> ### Author Response · Authors · 2024-11-26
> **Comments on the weaknesses you pointed out**
>
> Thank you for your assessment. To comment on the weaknesses you pointed out:
>
> **Benchmarking**: The method referred to as MEEA can be understood as the empirical variant of nested kriging, a commonly used approach in surrogate modeling and regression tasks. Current model aggregation techniques are largely focused on regression scenarios involving surrogate models, which is why the Boston Housing dataset was included as a sanity check to demonstrate that our method can improve performance in such settings.
> The primary contribution of our work, however, lies in the seamless, non-intrusive, and versatile applicability of the proposed approach, which enables novel applications of model aggregation to PDE solvers and operator learning. These domains present unique challenges, such as aggregating pre-trained, heterogeneous, or black-box models, and lack established benchmarks or comparable methods. By addressing these complexities, our method provides a robust solution for real-world scientific computing tasks, demonstrating its capability to unify and enhance predictions across diverse legacy models for complex systems.  As observed by referee 3, `` The capacity to aggregate a variety of models, including those derived from different methodologies and domains, underscores the versatility of this approach. This has important implications for numerous fields where the integration of multiple predictive models is advantageous, such as climate modeling, where ensemble methods are employed but often rely on simplistic aggregation techniques.’’
>
> **Theoretical guarantees** : MEVA can be seen as MEEA with the added assumption of unbiased models, which introduces a level of regularization to the aggregation process. This trade-off between regularization and generalization performance is a well-known principle in machine learning. In our revised version, we prove a theorem showing that MEVA converges faster than MEEA, making it a more robust aggregation. As we operate in a low data regime (because N datapoints need to be extrapolated to all x), this faster convergence rate is key in making MEVA superior to MEEA.
> Importantly, in all our experiments, MEVA consistently outperformed MEEA, providing strong empirical evidence for its superiority.
> > **Remark**: simple averaging is also an unbiased estimator (as coefficients sum to 1), thus it is automatically inferior to MVA.
>
> **Hyperparameter tuning**: Our method is formulated as a supervised learning problem, and standard approaches for hyperparameter selection, such as cross-validation or evaluation on a separate test set, are applicable here. To aid reproducibility, we have also specified the hyperparameter choices used in our experiments in the code, which is publicly available on the paper’s GitHub repository.

---

> > ### Comment · Reviewer_k7LR · 2024-11-26
> > **Benchmarking**
> >
> > I appreciate the authors' responses; however, I remain unconvinced by the response to my earlier comment regarding the lack of clear evidence that this paper advances the current state-of-the-art. The work would have been far more compelling if the authors had explored a more seriously developed example (e.g., ensemble methods for climate models as the authors mentioned). While I understand that there is no time to address this now, I believe that relying on a sanity check with the Boston housing dataset seems insufficient. The PDE solvers problem looks much more interesting though.

---

> > > ### Author Response · Authors · 2024-11-27
> > >
> > > Thank you for your response. We appreciate your comments.
> > >
> > > **State-of-the-art clarification**: Please note that the state-of-the-art is problem-dependent. For tasks where training data is accessible, such as the Boston housing dataset example, our method is competitive with established techniques. This example serves primarily as a sanity check, demonstrating that MEVA performs well in data science contexts, even when competing with methods specifically optimized for such tasks.
> > >
> > > **Advancing state-of-the-art in black-box settings**: For problems involving black-box models, where access to training data is unavailable or models are heterogeneous, MEVA represents the current state-of-the-art. The aggregation of PDE solvers exemplifies this, offering a clear benchmark with computable errors. Such settings are common in Computational Sciences, Engineering, and National Labs, where legacy models often require non-intrusive aggregation methods.
> > >
> > > **Broader impact**: While climate modeling and similar high-profile applications would indeed showcase MEVA’s potential further, such explorations are planned for future work. For this paper, we focused on providing a rigorous foundation and illustrative examples that highlight MEVA’s versatility and effectiveness across problem types.

---

> > > > ### Comment · Reviewer_k7LR · 2024-11-27
> > > >
> > > > I have had one more look at the experimental results for the aggregation of PDE solvers and am now more convinced that the results reported in tables 1 and 2 can be viewed as solid evidence supporting the proposed method in the paper. Consequently, I have adjusted my evaluation to a grade of 6 (marginally above the acceptance threshold).

---

> ### Author Response · Authors · 2024-11-26
> **To answer your questions**
>
> - In equation 13, we use for the Boston housing dataset a Gaussian process with Matern kernel (equation 13), and for the PDE example, the Fourier Neural Operator (lines 427 to 431)
>
> - Numerical solvers for PDEs, such as Laplace and Burger’s equations, often achieve machine precision within reasonable computational timeframes. In contrast, machine learning approaches, including operator learning methods like FNO, DeepONet, and kernel-based methods, typically struggle to surpass 1% accuracy in general settings. This creates an interesting challenge for machine learning methods to aggregate models that operate with orders of magnitude greater precision than direct approximation methods.
> Moreover, operator learning problems are inherently infinite-dimensional tasks, making them much closer to scientific computing challenges seen in fields like climate modeling or material science. This highlights the relevance of our work in addressing such problems, where the need to combine highly precise models with varying strengths becomes critical.
>
>
> - Obtaining multiple predictions for a single quantity of interest is undoubtedly more computationally expensive than relying on the best individual model. However, in many practical settings, such as weather prediction, ensembles are widely used because accuracy is prioritized over computational cost.
> The trade-off between computational cost and accuracy is context-dependent and can only be evaluated in practice, where specific constraints and requirements are known. In our experiments, most examples were completed within a few seconds and demonstrated an order-of-magnitude improvement in error, which justifies the additional computational expense in these cases.

---

> ### Author Response · Authors · 2024-11-26
> **New version of the manuscript**
>
> We have made several changes to the manuscript to improve the article.
>
> We have modified the title, abstract, and introduction to better highlight the contributions of our work. In particular, we have revised the related work section and added a clear contributions section.
> We have proved a theorem to address concerns about the well-foundedness of the method. This theorem justifies the superiority of MEVA over MEEA by showing it has a faster convergence rate, making it more robustly estimated. We identify our setting as a low data regime, making robust estimation key for accuracy.
> We have reworked the Boston housing dataset example. Its role as a sanity check for the method has been emphasized. We also noticed the small data set size made results highly variable. We now average results over many seeds, and after better hyperparameter choice, our method shows a 2% overall improvement.

---

### Official Review · Reviewer_E4o3 · 2024-11-09

**Soundness:** 3
**Presentation:** 4
**Contribution:** 2
**Rating:** 6
**Confidence:** 3

**Summary:**

In this paper, the authors study the problem of model aggregation where one intends to learn to combine the predictions given by different models. The authors study two variants, Minimal empirical error aggregation and Minimal empirical variance aggregation, both of which differ mainly with respect to the loss functions used. The former penalizes the error w.r.t. the aggretage function, while the latter takes into account the errors of the individual models. The authors study two applications: aggregating models in housing price prediction and learning aggregated solvers for PDE, the motive being illustrating that the MEVA learns to pick the model with low error for any given input.

**Strengths:**

- The paper is written well, easy to follow. The key ideas are articulated nicely.
- The contributed solutions seem significant more in the context of PDE solvers, where the metrics for the aggregated model outperform that of the baselines.

**Weaknesses:**

It is not very clear in the data-science settings when the aggregated models would be useful. Few more experiments migt add more clarity to that. For instance, the ensemble methods such as random forests achieve similar performance to that of the propsoed schemes. Would there be any advantage in using these aggregate schemes upon the existing benchmarks in data-science problems ?

While the proposed formulations differ mainly with respect to the loss function, what may be the data assumptions which would help us understand the performance differences of one loss over other better ? Specifically, when do we expect the loss function (5) to be better than (13) and vice versa ? Some analysis w.r.t. the underlying data assumptions may help understanding them better.

**Questions:**

Would learning the functions in the pathological examples be valid if the input samples are chosen randomly rather than in the specific patterns ? Does this mean that the failure of learning the underlying model is partly due to the choice of the points which are presented for learning ?

It was mentioned that MEVA doesn't improve the performance in settings when P not equals Identity, (line 263), what is the implication of that in practical cases ? How practical is the assumption that expect multiple models' errors to be independent ?

---

> ### Author Response · Authors · 2024-11-26
> **Comments on the weaknesses you pointed out**
>
> Thank you for your assessment. To comment on the weaknesses you pointed out:
>
> **The data science setting**: The proposed method is most effective in scenarios involving **heterogeneous models**, where traditional ensemble methods like random forests or gradient boosting are not directly applicable. Unlike these approaches, which are designed for homogeneous, trainable models in structured data settings, MEVA operates seamlessly on pre-trained **black-box models**, requiring no access to their internal training processes or data. This flexibility makes MEVA particularly valuable in **structured problems with inherent design constraints**, such as scientific computing, multi-physics simulations, and domain adaptation. In these contexts, models often include non-trainable numerical solvers, legacy simulation codes, or specialized machine learning models with distinct trade-offs that cannot be easily retrained or integrated using standard ensemble techniques. By dynamically adapting to heterogeneous model errors and minimizing variance, MEVA ensures robustness to poor-performing models and excels in aggregating specialized or black-box models, offering a versatile solution for challenging aggregation tasks.
>
> While data science tasks provide a useful validation context, they are not the primary domain where MEVA delivers the most practical value. Ensemble methods like boosting and random forests are highly effective in these settings, leveraging homogeneous, trainable models to achieve state-of-the-art performance. MEVA is not intended to compete with these established methods but instead addresses problems where standard approaches fail, such as reconciling the limitations of pre-trained or specialized models in **constraint-heavy, structured problems**. For instance, MEVA excels in aggregating heterogeneous models trained on different data subsets or domains, combining outputs from models with vastly different structures, or leveraging a mix of machine learning models and numerical solvers. While our experiment on the Boston Housing dataset serves as a sanity check to validate the method on a widely recognized benchmark, MEVA's core contributions and applicability lie in **scientific computing and real-world aggregation scenarios** where traditional ensemble techniques are insufficient.
>
> **Data Assumptions**: Our work focuses on developing a robust aggregation method under the assumption that the data is given. While the quality and diversity of data undeniably influence performance, as in any data-driven computation, our primary contribution lies in addressing aggregation challenges rather than introducing novel insights into data requirements beyond standard machine learning principles. That said, MEVA’s regularization through the unbiasedness assumption makes it particularly effective in **scarce data regimes**, where overfitting or error compounding in traditional methods often leads to degraded performance. Our new theorem proves that MEVA converges faster than MEEA, making it a more robust aggregation.

---

> ### Author Response · Authors · 2024-11-26
> **To answer your questions**
>
> **Pathological Examples and Input Distribution**: The pathological example indeed relies on specific point placements to illustrate potential pitfalls of naive aggregation methods like MEEA. It is a deliberately extreme scenario designed to highlight how a method may overfit by interpolating data rather than learning a robust aggregation. While such specific patterns may not always occur in practice, the broader takeaway is that methods overly reliant on empirical error minimization (like MEEA) can fail in scenarios with structured or biased sampling. This extreme vulnerability is validated by our Gaussian Process (GP) computations, theorem, and experimental results, which consistently show MEEA performing poorly under less extreme conditions: when input samples are randomly distributed, the likelihood of extreme cases diminishes, but the underlying risks of overfitting or poor generalization remain for MEEA, particularly in data-scarce or heterogeneous settings.
>
> **Implications of Independence Assumptions** : Allowing $P\neq I $ adds flexibility by enabling MEVA to account for correlated errors between models. However, our experiments show that this flexibility is often unnecessary, as independent errors are typically sufficient for robust performance. In practice, model independence is a reasonable assumption when models are developed using distinct assumptions, methods, or datasets. Even models trained on the same dataset (e.g., linear regression,  SVR, boosting) often exhibit uncorrelated generalization errors due to their differing inductive biases and decision boundaries. When errors are significantly correlated, MEVA with $P\neq I $ can theoretically improve performance, but this comes at the cost of increased complexity and potential sensitivity to estimation inaccuracies in $P$. In our tests, the added flexibility did not lead to significant performance gains, making $P=I$ a practical and reliable default for most scenarios.

---

> ### Author Response · Authors · 2024-11-26
> **New version of the manuscript**
>
> We have made several changes to the manuscript to improve the article.
>
> - We have modified the title, abstract, and introduction to better highlight the contributions of our work. In particular, we have revised the related work section and added a clear contributions section.
> - We have proved a theorem to address concerns about the well-foundedness of the method. This theorem justifies the superiority of MEVA over MEEA by showing it has a faster convergence rate, making it more robustly estimated. We identify our setting as a low data regime, making robust estimation key for accuracy.
> - We have reworked the Boston housing dataset example. Its role as a sanity check for the method has been emphasized. We also noticed the small data set size made results highly variable. We now average results over many seeds, and after better hyperparameter choice, our method shows a 2% overall improvement.

---

### Author Response · Authors · 2024-11-26
**New version of the manuscript**

We want to thank the reviewers for their constructive comments. We have made several changes to the manuscript to improve the article.

- We have modified the title, abstract, and introduction to better highlight the contributions of our work. In particular, we have revised the related work section and added a clear contributions section.
- We have proved a theorem to address concerns about the well-foundedness of the method. This theorem justifies the superiority of MEVA over MEEA by showing it has a faster convergence rate, making it more robustly estimated. We identify our setting as a low data regime, making robust estimation key for accuracy.
- We have reworked the Boston housing dataset example. Its role as a sanity check for the method has been emphasized. We also noticed the small data set size made results highly variable. We now average results over many seeds, and after better hyperparameter choice, our method shows a 2% overall improvement.

---

### Meta-Review · Area_Chair_yHwN · 2024-12-24

**Metareview:**

This paper introduces Minimal Empirical Variance Aggregation (MEVA), a novel framework for combining predictions from diverse models, treated as black boxes, in a way that enhances robustness and accuracy. The problem it addresses is the limitation of existing aggregation methods, like averaging or Minimal Empirical Error Aggregation (MEEA), which can overfit and fail in data-scarce or noisy scenarios. MEVA shifts the focus from error minimization to variance minimization, using a probabilistic framework to derive aggregation weights that ensure unbiasedness and robustness. Unlike traditional methods, MEVA seamlessly integrates heterogeneous models, including machine learning algorithms and numerical solvers, without requiring access to their internal mechanics. Its novelty lies in theoretical guarantees of faster convergence, better performance in scarce data regimes, and its application to complex, structured problems, such as solving partial differential equations (PDEs) and other operator learning tasks.

Core claims seems to be that MEVA outperforms MEEA in combining models by focusing on reducing prediction variability (variance) rather than directly minimizing error, making it more reliable in data-scarce or noisy settings. The theoretical results show that MEVA’s loss converges to its optimal value at a faster rate of $O(1/N)$, while MEEA converges more slowly at $O(1/\sqrt{N})$. This faster convergence happens because MEVA is less sensitive to noise and data imperfections, providing a more stable and effective way to balance the contributions of different models.

I would recommend the authors to add a synthetic core takeaway in the contributions sections.

**Additional Comments On Reviewer Discussion:**

The reviewers highlight the novelty of the paper as a robust model aggregation framework that shifts focus from error minimization to variance minimization, making it particularly effective for heterogeneous and black-box models. They commend the theoretical contributions, especially the proof of MEVA's faster convergence rate over MEEA, and its practical application to aggregating PDE solvers. However, they also raise concerns about limited experimentation, particularly in data science contexts where traditional ensemble methods like random forests excel. Some reviewers request a clearer comparison to existing aggregation benchmarks, such as econometrics model averaging or contemporary ensemble methods, and additional exploration of MEVA's handling of biased models or correlated errors. Despite these critiques, they agree on the method's strengths, particularly its applicability in scientific computing and structured problems, and consider the theoretical and empirical contributions sufficient for acceptance, albeit with room for broader application demonstrations in future work.

---

### Decision · Program_Chairs · 2025-01-22

Accept (Poster)